# A Simple Efficiency Incremental Learning Framework via Vision-Language Model with Multi-Adapters

## Abstract

Incremental Learning (IL) aims to learn new tasks while preserving previously acquired knowledge. Integrating the zero-shot learning capabilities of pre-trained vision-language models into IL methods has marked a significant advancement. However, these methods face three primary challenges: (1) the need for improved training efficiency; (2) reliance on a memory bank to store previous data; and (3) the necessity of a strong backbone to augment the model's capabilities. In this paper, we propose **SimE**, a **Sim**ple and **E**fficient framework that employs a vision-language model with an adapter designed specifically for the IL task. We report a remarkable phenomenon: there is not always a direct positive correlation between the number of adaptive adapter connections and the model's IL capabilities. While increasing the number of adapter connections between transformer blocks positively impacts model performance, adding more adaptive connections within transformer blocks during smaller incremental steps does not enhance, and may even degrade the model's IL ability. Such improvements only occur at more advanced incremental stages. Extensive experimental results show that SimE surpasses traditional methods by 9.6% on TinyImageNet and outperforms other CLIP-based methods by 5.3% on CIFAR-100. Notably, SimE, with only thousands of parameters and no memory bank, outperforms ZSCL, which has 140 million parameters, and surpasses CoOP, which requires a memory bank of size 1000. Furthermore, we conduct a systematic study to enhance the utilization of the zero-shot capabilities of CLIP. We suggest that the backbone encoder in SimE should use the image encoder from CLIP pre-trained on larger datasets, such as LAION-2B, and larger model architectures, such as ViT-L/14, for IL tasks.

## 1 Introduction

Deep learning models have achieved significant success when fully trained on domain-specific tasks. However, in real-world scenarios, new data often come from diverse sources. Training a deep learning model on such new data typically leads to the model forgetting previously learned information—a phenomenon known as catastrophic forgetting (Goodfellow et al., 2013). To address this issue, Incremental Learning (IL) methods have been proposed, drawing inspiration from the human ability to learn continuously. These methods enable models to preserve existing knowledge while acquiring new skills (De Lange et al., 2021; Masana et al., 2022). Traditional IL approaches, which start training from scratch (Li & Hoiem, 2017; Serra et al., 2018; Rebuffi et al., 2017), fail to leverage the zero-shot learning capabilities of pre-trained vision-language models. For example, Contrastive Language-Image Pre-training (CLIP) models (Radford et al., 2021), trained on extensive datasets, exhibit strong feature extraction abilities. Consequently, integrating CLIP's zero-shot learning capabilities into continual learning approaches has become a subject of keen interest (Thengane et al., 2022; Ding et al., 2022; Zheng et al., 2023; Zhou et al., 2022; Wang et al., 2023; Yu et al., 2024).

Despite the success of recent CLIP-based IL methods, several challenges remain. For example, the CoOP framework (Zhou et al., 2022) preserves historical knowledge by utilizing a memory bank that is periodically accessed and updated during IL tasks. However, the ever-expanding volume of accumulated data can overburden the memory bank, thereby constraining CoOP's capacity for lifelong learning. In contrast, Continual CLIP (Thengane et al., 2022) leverages a frozen pre-trained

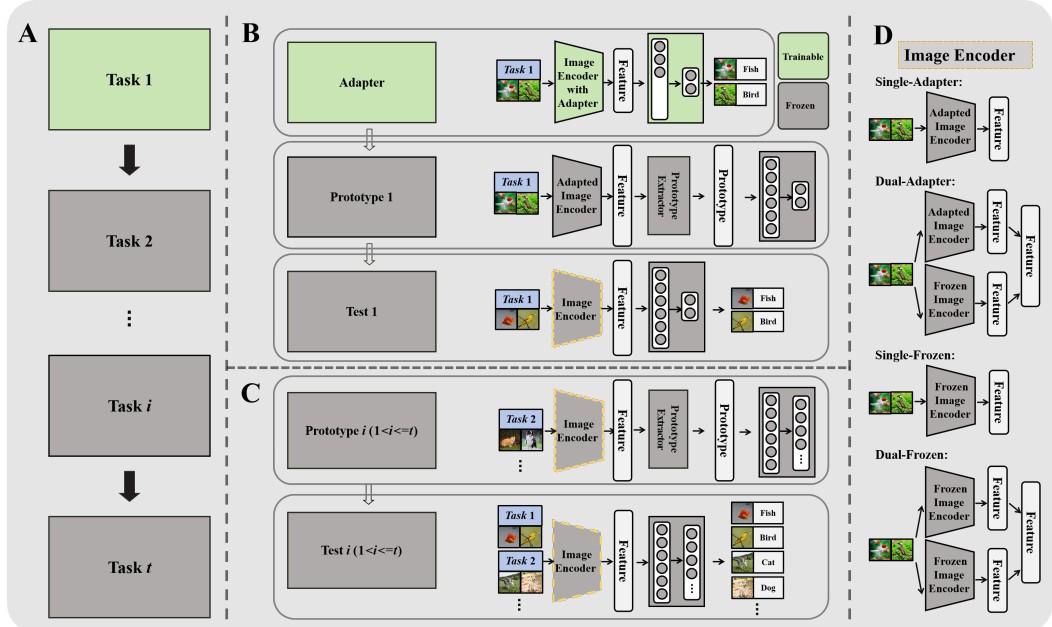

Figure 1: **The overall framework of SimE.** The green represents trainable and the grey denotes frozen components. **A**) illustrates the incremental learning tasks, which include t tasks. Specifically, we finetune the trainable parameters in SimE for task 1, while freezing all the parameters in SimE for the remaining tasks. **B**) The learning process for Task 1 can be divided into three stages: in the Adapter stage, the image encoder is finetuned using adapters; in the Prototype 1 stage, prototypes are computed based on the finetuned image encoder, and the classifier is updated; in the Test 1 stage, the classification performance of the model is evaluated. **C**) In the computation process for subsequent tasks $i$ ($1 < i < t$), all weights are frozen, only the prototypes are computed, and the classifier is updated. **D**) depicts the architectures of various image encoders.

CLIP to facilitate the model's continual learning capability, eliminating the need for replay memory but also limiting CLIP's zero-shot capabilities. Additionally, ZSCL (Zheng et al., 2023) employs parameter regularization through knowledge distillation to maintain the model's performance across IL tasks. However, ZSCL is not entirely efficient for IL endeavors, as it requires a substantial number of finetuning parameters to learn new data features and demands significant GPU resources during training. Moreover, the efficacy of CLIP's feature extraction is significantly influenced by the pre-trained datasets and the size of its backbone architecture (e.g., ViT). Despite this, there has been no systematic exploration into optimizing CLIP's zero-shot learning potential, which also affects the performance of IL methods. Collectively, the main challenges faced by these approaches include: **(1) the need for enhanced training efficiency**; **(2) reliance on a memory bank to store previous data**; and **(3) the need for a robust backbone to enhance the model's capabilities**.

To address these challenges, we introduce **SimE** (see Fig.1), a Simple and Efficient IL framework that combines a vision-language model with an adapter designed for efficient IL tasks. The adapter (Houlsby et al., 2019; Chen et al., 2022) is a lightweight module inserted into transformer blocks, enabling finetuning of the pre-trained model using minimal parameters. During training, the pre-trained model's parameters are frozen; we finetune only the adapter's trainable parameters, enhancing the model's parameter efficiency and adaptability without requiring a memory bank. We conduct a comprehensive evaluation of various backbones and pre-trained datasets to ascertain the most effective CLIP configurations for IL tasks using SimE. CLIP offers a spectrum of backbones, ranging from base to large models, as described by Radford et al. (2021), each with its own set of parameters. Additionally, the scope of pre-trained datasets is vast, as evidenced by works such as Gadre et al. (2024) and Cherti et al. (2023), which span from 400 million to 2 billion samples. Our systematic investigation delves into the influence of these disparate backbones and pre-trained datasets on the performance of CLIP in IL scenarios.

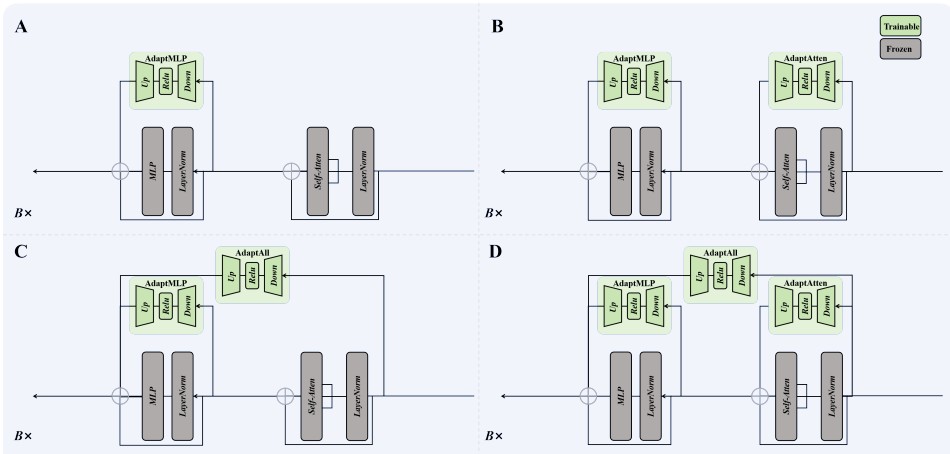

Figure 2: **Comparison of previous and current finetuning approaches: The previous approach, AdaptFormer (A), is contrasted with our Multi-Adapter finetuning (B, C, and D).** The modules colored in green are trainable, while those in gray are frozen. In AdaptFormer and Multi-Adapter, the AdaptMLP, AdaptAtten, and AdaptAll modules are parameterized by a bottom-up bottleneck module with trainable parameters, whereas the original MLP and Self-Attention modules remain frozen. The AdaptFormer consists of the original frozen branch coupled with AdaptMLP. In contrast, our Multi-Adapter incorporates various trainable modules alongside the frozen branch for enhanced adaptability. And $B\times$ is represented by $B$ Blocks.

In SimE, by simply combining CLIP and AdaptFormer, we observe that increasing the number of adapters between transformer blocks can improve model performance. To explore better methods of adapter connections, we propose a new adapter design named Multi-Adapter (see Fig.2), which aims to increase the number of adaptive connections beyond the constraints imposed by the AdaptFormer architecture. Surprisingly, we find that within transformer blocks, increasing the number of adaptive connections in smaller incremental steps does not enhance, and may even degrade the model's IL capabilities. This improvement only occurs in larger incremental stages. Extensive experiments across various settings demonstrate the effectiveness of SimE on IL tasks. Our contributions can be summarized as follows:

- We introduce SimE, which surpasses existing baseline IL models in class-incremental learning tasks. SimE is distinguished by its efficiency in three key areas: GPU usage, the number of trainable parameters, and memory size (as illustrated in Fig.5). Furthermore, SimE achieves competitive or superior accuracy with fewer additional parameters compared to other methods leveraging pre-trained models. (as shown in Fig.4(a)).

- We propose Multi-Adapter to explore better methods of adapter connections and observe a significant phenomenon: there is not always a direct positive correlation between the number of adaptive connections and the model's IL capabilities. While increasing the number of adapter connections between transformer blocks positively impacts model performance, within transformer blocks, adding more adaptive connections in smaller incremental steps does not enhance, and may even degrade the model's IL ability. Such improvement only occurs at more advanced incremental stages.

- We conduct a systematic study to enhance the utilization of the zero-shot capabilities of CLIP under SimE, pinpointing the most suitable backbone for CIFAR-100 and TinyImageNet. We advocate for the use of CLIP models that have been pre-trained on expansive datasets, such as LAION-2B, and possess larger architectures like ViT-L/14, to facilitate IL processes via SimE.

## 2 RELATED WORK

**Conventional Continual Learning Methods:** Traditional continual learning methods can be divided into three categories: regularization-based, architecture-based, and replay-based approaches. Regularization-based methods (Aljundi et al., 2018; Kirkpatrick et al., 2017; Li & Hoiem, 2017) mitigate forgetting by incorporating regularization terms into the loss function, encouraging the model to retain weights important for previous tasks. However, these methods may diminish the model's ability to learn new categories effectively. Architecture-based methods (Mallya & Lazebnik, 2018; Serra et al., 2018; Wang et al., 2020) adjust the network's structure to accommodate new tasks by expanding it or altering its configuration. While effective, these methods may not be ideal for task-agnostic continual learning and can lead to increased memory usage. Replay-based methods (Rebuffi et al., 2017; Buzzega et al., 2020; Cha et al., 2021) involve storing and periodically revisiting data from previous tasks to help the model retain prior knowledge. Although useful, these methods can raise privacy concerns and may be less effective with smaller data buffers. Moreover, traditional continual learning models are typically trained from scratch, which may limit the maximum achievable performance by not leveraging pre-trained models.

**Continual Learning Methods Using CLIP:** Recently, pre-trained models have been increasingly adopted in continual learning due to their powerful feature extraction capabilities (Wang et al., 2022c;b; Thengane et al., 2022). CLIP (Radford et al., 2021), renowned for its impressive zero-shot abilities, excels in feature extraction through contrastive learning on vast amounts of image-text pairs. Consequently, several studies (Thengane et al., 2022; Ding et al., 2022; Zheng et al., 2023; Zhou et al., 2022; Wang et al., 2023; Yu et al., 2024) have integrated CLIP into continual learning models to enhance performance. Continual-CLIP (Thengane et al., 2022) directly applies CLIP to continual learning without any finetuning, maintaining CLIP's feature extraction capacity but potentially suffering from domain gaps between pre-trained datasets and downstream tasks. LwF-VR (Ding et al., 2022) and ZSCL (Zheng et al., 2023) finetune the entire model using traditional continual learning methods to adapt to specific tasks. This process is computationally expensive due to the large size of pre-trained models and may also lead to the forgetting of previously learned knowledge. Thus, the finetuning strategy significantly impacts model performance.

**Continual Learning Methods Using Adapter Finetuning:** Adapters were initially introduced in natural language processing (Houlsby et al., 2019) to finetune pre-trained models for specific tasks by modifying a minimal set of weights. This approach has gained traction across various fields due to its notable efficiency (Chen et al., 2022; Dong et al., 2024). In the realm of continual learning, recent studies (Liu et al., 2023; Ermis et al., 2022b;a; Yu et al., 2024) have explored integrating adapters, placing them after the encoder or within the model's blocks. These adapters enable learning new tasks with a limited number of trainable parameters while preserving the core feature extraction functions. AdaptFormer (Chen et al., 2022), known for its lightweight parameterization, enhances model efficiency but is limited by the number of adaptive connections it can establish. In this paper, we introduce a Multi-Adapter that expands the number of adaptive connections, thereby extending the model's flexibility.

## 3 THE SIME FRAMEWORK VIA VISION-LANGUAGE MODELS WITH ADAPTERS

In this section, we first introduce the definition of the incremental learning (IL) task. Next, in Section 3.1, we present SimE, a framework that combines the image encoder in vision-language models with an adapter. Then, we introduce the formulation of the Multi-Adapter in Section 3.2. Finally, in Section 3.3, we describe implementations of SimE using the image encoder from CLIP and the adapters from AdaptFormer and Multi-Adapter. Incremental learning (IL) methods enable a model to learn new tasks while retaining knowledge from previous ones.

Consider a sequence of tasks $\mathcal{D} = D_1, D_2, \ldots, D_T$, where the $t$-th task is defined as $D_t = (\mathbf{x}_i^t, \mathbf{y}_i^t)i = 1^{m_t}$. Here, $D_t$ contains $m_t$ samples $\mathbf{x}_i^t$ and their corresponding labels $\mathbf{y}_i^t$. During the learning of task $D_t$, we have access only to the data from $D_t$; the data from previous tasks $D_1, D_2, \ldots, Dt - 1$ are unavailable. Furthermore, we focus on task-agnostic class-incremental learning (class-IL), where historical data cannot be used for rehearsal, and the task ID is not known during inference. In this setting, the model must learn to classify samples from all classes seen so far without explicit information about which task a sample belongs to.

## 3.1 SimE formulation

SimE is structured into three primary phases: data pre-processing, feature extraction, and image classification. In the initial phase, raw image data are transformed into a format compatible with the model's requirements. This is followed by the feature extraction phase, where an encoder—specifically the image encoder from a pre-trained vision-language model equipped with an adapter and prototype extractors—processes the formatted images. The process culminates in the image classification phase, where a fully connected (FC) layer acts as the classifier. This classifier is intricately designed to support class-incremental learning (class-IL), facilitating the seamless incorporation of new classes.

**The Encoder.** Image encoders are utilized to extract visual features from preprocessed images. Commonly used image encoder architectures include ResNet and ViT. Taking ViT as an example, the $i$th block of basic transformer module can be described as follows: 1) self-attention $f_i : \mathcal{X}_i \to \mathcal{A}_i$, which computes the attention among elements and learns global information through their interactions; 2) MLP $g_i : \mathcal{A}_i \to \mathcal{H}$, which applies nonlinear transformations to the input sequence to enhance the model's expressive capacity. Formally, this can be represented as:

$$i \text{ th Self-Attention: } \boldsymbol{a}_i = f_i(\theta_i, \boldsymbol{x}_i), \ i \text{ th MLP: } \boldsymbol{h}_i = g_i(\phi_i, \boldsymbol{a}_i). \tag{1}$$

Here, $\boldsymbol{h}_i$ contains the visual features of the original image $\boldsymbol{x}_i$, Self-Attention and MLP are instantiated with the parameters $\theta_i$ and $\phi_i$ respectively. For the pre-trained encoder, both $\theta_i$ and $\phi_i$ are pre-trained weights that are frozen.

**The Adapter.** An adapter is a lightweight module designed to finetune pre-trained models for downstream datasets with a minimal number of additional parameters. The parameters of adapters are trainable and will be updated during the finetuning process, while the weights of pre-trained models are frozen. The adapter in the $i$th blocks extracts features as $d_i : \mathcal{X}_i \to \mathcal{H}_i$,

$$i \text{ th Adapter: } \boldsymbol{h}_i = d_i(\tilde{\eta}_i, \boldsymbol{x}_i). \tag{2}$$

Here, adapter $i$ is instantiated with parameters $\tilde{\eta}_i$, where $\tilde{\eta}_i$ are trainable. The visual features extracted by the adapter are integrated into the pre-trained encoder, enhancing its ability to extract visual features of downstream datasets. It is noteworthy that, unlike the modules of the pre-trained encoder, both the number and positions of adapters are variable. By introducing adapter into pre-trained encoder, we get the general form of the encoder with adapter $E(\boldsymbol{x})$:

$$E(\boldsymbol{x}) = \sum_i^B (g_i(\phi_i, f_i(\theta_i, \boldsymbol{x}_i)) + d_i(\tilde{\eta}_i, \boldsymbol{x}_i)), \tag{3}$$

where $B$ is the number of the block in the encoder. Especially, when $i = 0$, the $\boldsymbol{x}_i$ is the reprocessed image $\boldsymbol{x}$ .

**The Prototype extractor.** In image classification, we follow Snell et al. (2017), setting the average features of the classes as the weights of the classifier. For the $t$-th task ($t = 2, \ldots, T$), we do not update the weights of $E(\boldsymbol{x})$ in Eq.3; instead, we use the $E(\boldsymbol{x})$ directly to calculate the average value of features and set it as prototypes in datasets $\{\boldsymbol{x}_i^1, \ldots, \boldsymbol{x}_i^t\}_{i=1}^{n_t}$:

$$p_k = \frac{1}{K} \sum_{j=1}^{|D^t|} I(y_j = k) E(\boldsymbol{x}), \tag{4}$$

where $p_k \in \mathbb{R}^d$ is the prototype of increment class $k$ in $t$-th task, $K = \sum_{j=1}^{||D^t||} I(y_j = k)$, $I(\cdot)$ is the indicator function. $p_k$ contains the average features of class $k$, implying that the images of class $k$ should exhibit the greatest similarity with $p_k$ among all prototypes.

**The Classifier**. In Class IL, the classifier is dynamic and can be implemented in various ways(Mai et al., 2021; Wang et al., 2023). In this paper, we use a FC layer as our classifierSnell et al. (2017). For the $t$-th task, the classifier is an FC layer $W_t \in \mathbb{R}^{D \times (N+M)}$, where $D$ is the feature dimension, $N$ is the number of classes at $(t\text{-}1)$-th task, $M$ is the number of increment classes in $t$-th task. We use the training set data $\boldsymbol{x}_{\text{pro}} = \{(\boldsymbol{x}_i^t, \boldsymbol{y}_i^t)\}_{i=1}^{n_t}$ from the $t$-th task to calculate the prototype $W_{\text{pro}} = \text{mean}(E(\boldsymbol{x}_{\text{pro}}))$, where $n_t$ is the number of samples in the training set of the $t$-th task, then update the FC layer $W_t$: $W_t = W_{t-1} + W_{\text{pro}}$, where $W_{t-1} \in \mathbb{R}^{D \times N}$ and $W_{\text{pro}} \in \mathbb{R}^{D \times M}$. The

cosine similarity for classification is then calculated as:

$$f(\boldsymbol{x}) = \left(\frac{\mathbf{W}}{\|\mathbf{W}\|_2}\right)^\top \left(\frac{E(\boldsymbol{x})}{\|E(\boldsymbol{x})\|_2}\right) \tag{5}$$

given that prototype $p_i$ is most similar to instances of class $i$, it is expected that the classifier will assign a higher probability to the correct class label.

## 3.2 Multi-Adapter formulation

We propose the Multi-Adapter, which comprises three adapter sub-modules: AdaptAtten, AdaptMLP, and AdaptAll as shown in Fig.2. The sub-modules AdaptAtten, AdaptMLP, and AdaptAll share the same structure, each containing a down-projection, a non-linear activation function (e.g., ReLU) and an up-projection. Thus, we can derive a more specific form of the $i$th adapter $r_i : \mathcal{C}_i \to \hat{\mathcal{S}}$,

$$\boldsymbol{s}_{ij} = r_{ij}(\tilde{\eta}_{ij}, \boldsymbol{c}_{ij}), \; where \; \boldsymbol{c}_{ij} = \left\{ \begin{array}{llll} \boldsymbol{a}_i & j = 1 \; and & \boldsymbol{s}_{ij} = \boldsymbol{h}_i; & r_{ij} \; is \; \text{AdaptMLP} \\ \boldsymbol{x}_i & j = 2 \; and & \boldsymbol{s}_{ij} = \boldsymbol{a}_i; & r_{ij} \; is \; \text{AdaptAtten} \\ \boldsymbol{x}_i & j = 3 \; and & \boldsymbol{s}_{ij} = \boldsymbol{h}_i; & r_{ij} \; is \; \text{AdaptAll} \end{array} \right. , \tag{6}$$

here $\boldsymbol{c}_{ij}$ can be equal to the initial input $\boldsymbol{x}_i$ or an intermediate variable $\boldsymbol{a}_i$, in Eq.1, and $j \in \{1, 2, 3\}$, correspond to AdaptMLP, AdaptAtten, and AdaptAll, respectively. Thus, the $i$ th block in ViT can be represented as a combination of the pre-trained modules and the adapter sub-modules:

$$E'(\boldsymbol{c}) = \sum_i^B \sum_j^Z (f_{ij}(\theta_{ij}, g_{ij}(\phi_{ij}, \boldsymbol{c}_{ij})) + r_{ij}(\tilde{\eta}_{ij}, \boldsymbol{c}_{ij})), \tag{7}$$

where $B$ is the number of the block in the encoder and $Z$ is a subset of $\{1, 2, 3\}$ ($Z \subseteq \{1, 2, 3\}$). Especially, when $i = 0$, the $\boldsymbol{c}_{0j}$ is the reprocessed image $\boldsymbol{x}$. By identifying the trainable parameters, the adapter can be instantiated. Lastly, the optimisation of adapter for domain adaptation can see in Appendix A.

## 3.3 Realizations of SimE via CLIP with different adapters

In SimE, there are numerous implementations for the encoder and adapter. Here, we first employ the CLIP visual encoder as the encoder and AdaptFormer or Multi-Adapter as the adapter to establish a toy model of SimE. Subsequently, we explore various specific implementations of SimE. **CLIP** (Radford et al., 2021) is a powerful visual-language contrastive learning model comprising an image encoder and a text encoder. It is trained on 400 million image-text pairs and possesses strong feature extraction capabilities. In this paper, we employ the CLIP pre-trained visual encoder as our backbone, i.e., we instantiate our encoder using the CLIP pre-trained weights $\theta$ and $\phi$. Additionally, there exist CLIP models pre-trained on different datasets, corresponding to different encoder instances. It is noteworthy that during the continual learning process, the pre-trained weights of CLIP are frozen and do not participate in weight updates. **Adaptformer** Chen et al. (2022) containing a down-projection $W_{\text{down}} \in \mathbb{R}^{D \times R}$ to reduce the feature dimension, a non-linear activation function(ReLU) and an up-projection $W_{\text{up}} \in \mathbb{R}^{R \times D}$ to project the features back to their original dimension, where $D$ is the feature dimension and $R$ is the dimension of bottleneck. The specific form of the AdaptFormer is:

$$d_i(\tilde{\eta}_i, \boldsymbol{h}_i) = \alpha \text{ReLU}(\boldsymbol{h}_i \cdot W_{\text{down}}) \cdot W_{\text{up}} \tag{8}$$

Here $\alpha$ is the scaling factor in the residual connection, which is set to 0.1 by default in AdapterFormer. **Multi-Adapter** is represented by Eq.7. The Adaptformer is a special case in **Multi-Adapter** and we employ the same down-up projects in Eq.8 to initialize the Multi-Adapter.

## 4 Experimental results

In this section, we begin by comparing the performance of the proposed SimE method with that of other Class-Incremental Learning (CIL) methods. Next, we evaluate the efficiency of these models by examining their number of trainable parameters, GPU usage, and memory bank size. Furthermore, we conduct ablation experiments to investigate the impact of various components within SimE. Lastly, the details of the experimental settings are provided in Appendix B.

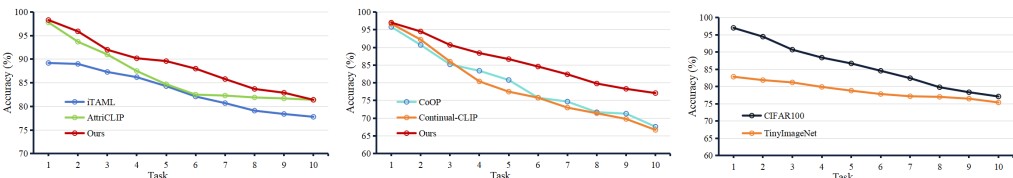

Figure 3: Last accuracy of every Task in 10 steps. The Last accuracy of Task $t$, $t \in \{1, 2, ..., 10\}$ is the Top-1 accuracy over all the previous Tasks (i.e., Tasks 1, 2, ..., $t$). The results in **left** and **middle** are conducted on CIFAR100. The result of "Ours" in **left** is based on ViT-L/14 and in **middle** and **right** are based on ViT-B/16.

## 4.1 COMPARISON ON THE ACCURACY OF DIFFERENT CIL METHODS

**SimE outperforms most Other CIL methods at various steps on both CIFAR-100 and Tiny-ImageNet datasets.** First, we compare the performance of our method with other CIL methods across all tasks and present the results in Tab.1, where the best results are highlighted in grey. The results, measured by the average accuracy across tasks, show that our method achieves the highest scores among recent state-of-the-art methods, demonstrating the significant effectiveness of incorporating adapters into a pre-trained model. Specifically, our method and other CLIP-based methods (CoOP(Zhou et al., 2022), Continual-CLIP(Thengane et al., 2022)) have a substantial advantage over traditional continual learning methods initially, reflecting the superior feature extraction capabilities of pre-trained models.

However, the accuracy of other CLIP-based methods drops quickly as training progresses, indicating that they are severely affected by domain gaps or catastrophic forgetting. Our method not only outperforms CLIP-based methods at the start, showing that finetuning helps the model adapt to downstream tasks, but also exhibits a slower decline in performance because it retains the original feature extractor, thus preserving the pre-trained model's prior knowledge. In addition to splitting CIFAR-100 into 10 tasks, we also experimented with 20 and 50 tasks and have listed the results in Tab.1. Our method consistently performs the best across all settings, outperforming current state-of-the-art methods by at least 3%. Furthermore, we conducted experiments on TinyImageNet, where we split the dataset into multiple tasks with 100 classes as base classes, and reported these results in Tab.1. Our method remains superior in most settings, further demonstrating its effectiveness.

Table 1: Comparison on the accuracy of different CIL methods. The Average and Last accuracy of different CIL methods on CIFAR100 and TinyImageNet benchmark. Among them, UCIR(Hou et al., 2019), PASS(Zhu et al., 2021), DyTox(Douillard et al., 2022), and DER(Yan et al., 2021) train from scratch, while the remaining methods use CLIP ViT-B/16 as the backbone, where † indicates the result based on the CLIP ViT-L/14 pre-trained on Laion-2B. The 100 classes of TinyImageNet are used as base classes. The best results are coloured grey.

| | CIFAR100 | | | | | | TinyImageNet | | | | | |
| | 10 Steps | | 20 Steps | | 50 Steps | | 5 Steps | | 10 Steps | | 20 Steps | |
| Methods | Avg | Last | Avg | Last | Avg | Last | Avg | Last | Avg | Last | Avg | Last |
|---|---|---|---|---|---|---|---|---|---|---|---|---|
| UCIR(Hou et al., 2019) | 58.66 | 43.39 | 58.17 | 40.63 | 56.86 | 37.09 | 50.30 | 39.42 | 48.58 | 37.29 | 42.84 | 30.85 |
| PASS(Zhu et al., 2021) | - | - | - | - | - | - | 49.54 | 41.64 | 47.19 | 39.27 | 42.01 | 32.93 |
| DyTox(Douillard et al., 2022) | 67.33 | 51.68 | 67.30 | 48.45 | 64.39 | 43.47 | 55.58 | 47.23 | 52.26 | 42.79 | 46.18 | 36.21 |
| DER(Yan et al., 2021) | 74.64 | 64.35 | 73.98 | 62.55 | 72.05 | 59.76 | - | - | - | - | - | - |
| CLIP(Radford et al., 2021) | 74.47 | 65.92 | 75.20 | 65.74 | 75.67 | 65.94 | 69.62 | 65.30 | 69.55 | 65.59 | 69.49 | 65.30 |
| Fien-tune | 65.46 | 53.23 | 59.69 | 43.13 | 39.23 | 18.89 | 61.54 | 46.66 | 57.05 | 41.54 | 54.62 | 44.55 |
| iCaRL(Rebuffi et al., 2017) | 79.35 | 70.97 | 73.32 | 64.55 | 71.28 | 59.07 | 77.02 | 70.39 | 73.48 | 65.97 | 69.65 | 64.68 |
| LwF(Li & Hoiem, 2017) | 65.86 | 48.04 | 60.64 | 40.56 | 47.69 | 32.90 | 60.97 | 48.77 | 57.60 | 44.00 | 54.79 | 42.26 |
| Continual-CLIP(Thengane et al., 2022) | 75.17 | 66.72 | 75.95 | 66.72 | 76.49 | 66.72 | 70.49 | 66.43 | 70.55 | 66.43 | 70.51 | 66.43 |
| LwF-VR(Ding et al., 2022) | 78.81 | 70.75 | 74.54 | 63.54 | 71.02 | 59.45 | 77.56 | 70.89 | 74.12 | 67.05 | 69.94 | 63.89 |
| ZSCL(Zheng et al., 2023) | 82.15 | 73.65 | 80.39 | 69.58 | 79.92 | 67.36 | 80.27 | 73.57 | 78.61 | 71.62 | 77.18 | 68.30 |
| SimE(Ours) | 85.94 | 77.10 | 85.67 | 76.61 | 84.16 | 73.88 | 79.35 | 75.37 | 79.32 | 75.37 | 79.29 | 75.37 |
| SimE(Ours)† | 91.66 | 86.03 | 92.27 | 86.64 | 91.64 | 85.35 | 86.47 | 83.33 | 86.41 | 83.33 | 86.39 | 83.33 |

We further compare the results of each task with traditional and CLIP-based CIL methods, as shown in Fig.3 & Tab.5, and our method consistently outperforms other methods across all tasks. Additionally, in Fig.3 & Fig.6, we compare the performance of each task on CIFAR100 and TinyImageNet, revealing differences in the generalization ability of the backbone across different datasets. Consequently,

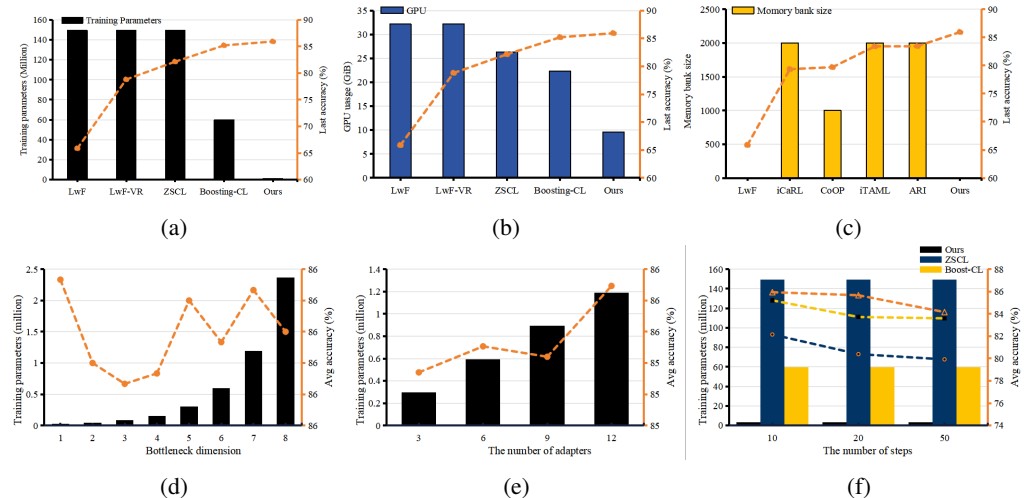

(a)      (b)      (c)

(d)      (e)      (f)

Figure 4: Comparison on the efficiency of different CIL methods. The dotted line and right axis coloured orange present the Last accuracy and Avg accuracy. (a)(b)(c) denote the Training parameters, GPU usage, Memory bank size and Last accuracy of different CIL methods respectively, (d)(e) is the Training parameters and Avg accuracy of Ours under different bottleneck dimensions and number of adapters. (f) show the comparison between Ours and other CIL methods in training parameters and Avg accuracy. All the experiments are conducted on CIFAR100, and (a)-(e) are conducted in 10steps.

we conduct further experiments to identify the optimal backbone for different datasets, as detailed in Sec. 4.4.

## 4.2 COMPARISON ON THE EFFICIENCY OF DIFFERENT CIL METHODS

We compare the efficiency of our proposed SimE method with other Class-Incremental Learning (CIL) methods by examining the number of trainable parameters, GPU usage, and replay data size. The experimental settings are the same as those described in Appendix B and the results are shown in Fig.4. As illustrated in Fig.4(a) and Fig.4, our method requires only thousands of trainable parameters while achieving competitive results compared to other CIL methods that utilize millions of parameters, significantly reducing training costs.Furthermore, as shown in Fig.4(b) and Fig.4(c), our method uses only one-third of the parameters and does not require a buffer to store replay data. We also study the influence of the bottleneck dimension and the number of adapters, as depicted in Figures Fig.4(d) and Fig.4(e). Despite varying these parameters, our method still achieves competitive performance with minimal trainable parameters. These results demonstrate that our method can achieve performance comparable to or even exceeding that of other CIL methods with minimal training costs, thereby strongly validating the efficiency of the proposed SimE method.

## 4.3 ABLATION STUDY ON THE INFLUENCE OF ADAPTER COMPONENTS IN SIME

**The influence of adapter connection between transformer blocks.** We investigated the impact of the position and number of adapters inserted between transformer blocks, presenting the results in Fig.5 & Tab.6 & Tab.7. In our notation, "1-3" indicates that adapters are inserted only into the first three blocks of the feature extractor. From Fig.5, it is evident that inserting the same number of adapters into the earlier blocks significantly improves model performance. This suggests that learning primary features plays a more crucial role in model finetuning. Additionally, we varied the number of adapters between transformer blocks from 0 to 12. Inserting adapters into every block (totaling 12 adapters) consistently yielded the best performance across all steps. Therefore, a larger number of adapters between transformer blocks leads to better model performance, indicating that increasing the number of adapter connections between transformer blocks positively impacts model outcomes.

**The influence of adapter connection within transformer blocks.** We also tested different implementations of the Multi-Adapter by inserting adapters within all 12 transformer blocks. The results

are reported in Tab.2 & Fig.7 & Tab.8 & Tab.9. Interestingly, we found that in smaller incremental steps, increasing the number of adaptive connections within transformer blocks does not improve model performance; in fact, it can even degrade it. The previously observed positive correlation only occurs in larger incremental steps. This suggests that a higher number of adapter connections within transformer blocks does not necessarily lead to better outcomes. In SimE, we explore the optimal implementation of adapters across various task configurations.

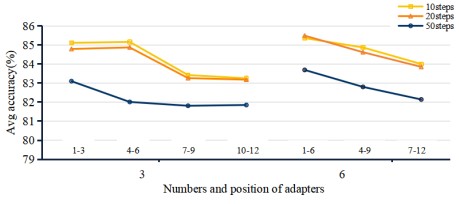 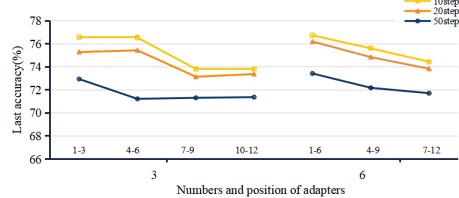

(a) Avg accuracy on different numbers of adapters     (b) Last accuracy on different numbers of adapters

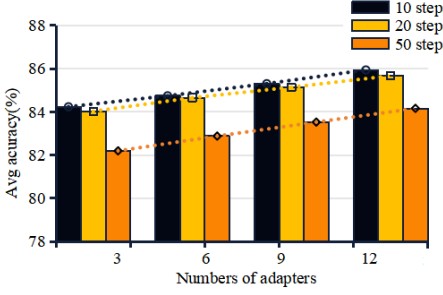 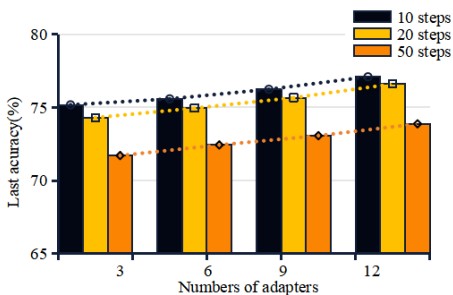

(c) Averaged Avg accuracy on different numbers of adapters     (d) Averaged Last accuracy on different numbers of adapters

Figure 5: Influence of adapters' position and number between transformer blocks. The x-axis represents the number of adapters in the encoder, with the numerical ranges indicating the positions of the adapters. For example, "1-3" signifies that adapters are inserted in the first 3 blocks. The accuracy shown in (c) and (d) represents the average results for different adapter positions with the same number of adapters. All results are based on CIFAR100.

Table 2: The results of different implementations of Multi-Adapter. The structures of Adapter-MLP, Adapter-Atten and Adapt-All are shown in Fig2. "Para" refers to trainable parameters, with "M" standing for million. All experiments are conducted on CIFAR100 and the best results are coloured grey

| Adapt-MLP | Adapt-Atten | Adapt-All | Para(M) | 10 steps Avg | 10 steps Last | 20 steps Avg | 20 steps Last | 50 steps Avg | 50 steps Last |
|---|---|---|---|---|---|---|---|---|---|
| ✗ | ✗ | ✗ | 0 | 79.69 | 70.08 | 80.41 | 70.08 | 80.80 | 70.08 |
| ✔ | ✗ | ✗ | 1.19 | 85.60 | 76.70 | 85.30 | 76.02 | 84.09 | 73.77 |
| ✗ | ✔ | ✗ | 1.19 | 85.94 | 77.10 | 85.67 | 76.61 | 84.16 | 73.88 |
| ✗ | ✗ | ✔ | 1.19 | 85.77 | 76.83 | 85.48 | 76.16 | 84.16 | 73.86 |
| ✔ | ✔ | ✗ | 2.38 | 85.73 | 76.79 | 85.42 | 76.09 | 84.76 | 74.66 |
| ✗ | ✔ | ✔ | 2.38 | 85.84 | 76.98 | 85.36 | 76.03 | 84.75 | 74.69 |
| ✔ | ✗ | ✔ | 2.38 | 85.63 | 76.65 | 85.12 | 75.66 | 84.75 | 74.76 |
| ✔ | ✔ | ✔ | 3.57 | 85.54 | 76.51 | 85.05 | 75.53 | 85.00 | 75.16 |

## 4.4 ABLATION STUDIES ON THE INFLUENCE OF CLIP COMPONENTS IN SIME

**The influence of pre-trained datasets.** CLIP has attracted significant attention due to its powerful zero-shot capabilities, leading many studies to retrain CLIP from scratch on other image-text pair datasets, such as Datacomp (Gadre et al., 2024) and LAION (Schuhmann et al., 2022). These datasets are comparable in size to or even larger than the original pre-training dataset (WIT-400M). We evalu-

ated their performance on CIFAR-100 and TinyImageNet. As shown in Tab.3, pre-training models on larger datasets generally enhances their feature extraction capabilities, leading to better generalization. However, there are instances where smaller pre-trained datasets yield higher accuracy, indicating that dataset quality and preprocessing techniques also significantly impact model performance.

Table 3: The influence of CLIP pre-trained datasets. WIT-400M is the closed-source dataset of OpenAI while others are from Open_CLIP. All results are conducted on ViT-B/16 and the 100 classes of TinyImageNet are used as base classes. The best results are coloured grey.

| | CIFAR100 | | | | | | TinyImageNet | | | | | |
| | 10 steps | | 20 steps | | 50 steps | | 10 steps | | 20 steps | | 50 steps | |
| Blocks | Avg | Last | Avg | Last | Avg | Last | Avg | Last | Avg | Last | Avg | Last |
|---|---|---|---|---|---|---|---|---|---|---|---|---|
| WIT-400M(Radford et al., 2021) | 85.60 | 76.70 | 85.30 | 76.02 | 84.09 | 73.77 | 79.35 | 75.37 | 79.32 | 75.37 | 79.29 | 75.37 |
| Laion-400M(Schuhmann et al., 2022) | 87.14 | 79.54 | 86.86 | 78.82 | 85.95 | 77.63 | 80.62 | 78.01 | 80.46 | 77.48 | 81.06 | 79.22 |
| Laion-2B(Schuhmann et al., 2022) | 88.34 | 81.33 | 88.47 | 80.89 | 87.91 | 80.09 | 81.98 | 79.99 | 81.83 | 79.77 | 82.78 | 81.63 |
| DataComp-1B(Gadre et al., 2024) | 88.04 | 80.77 | 87.89 | 79.88 | 87.57 | 79.25 | 81.38 | 79.11 | 81.49 | 78.70 | 82.51 | 80.89 |
| CommonPool-1B(Gadre et al., 2024) | 86.96 | 78.74 | 86.58 | 77.88 | 86.21 | 77.10 | 80.13 | 76.95 | 80.26 | 76.69 | 81.24 | 78.70 |

**The influence of ViT backbone size.** In addition to examining pre-trained datasets, we investigated the impact of different ViT backbones in CLIP. Our default model uses ViT-B/16. As shown in Tab.4, increasing the backbone size significantly improves model performance. Specifically, the accuracy of ViT-L consistently surpasses that of ViT-B across various settings, demonstrating superior feature extraction capabilities. When the model size is held constant, the size of the image patches plays a crucial role in feature extraction, with smaller patch sizes better capturing semantic information. In contrast, the size of the images during pre-processing has a relatively minor impact on model performance.

Table 4: The influence of CLIP ViT backbones size. The experiments use the corresponding data preprocessing while "336px" indicates the images are resized to 336. All experiments conducted on CIFAR100 and the best results are highlighted in grey.

| | 10 steps | | 20 steps | | 50 steps | |
| Blocks | Avg | Last | Avg | Last | Avg | Last |
|---|---|---|---|---|---|---|
| ViT-B/16 | 85.94 | 77.10 | 85.67 | 76.61 | 84.16 | 73.88 |
| ViT-B/32 | 83.60 | 74.43 | 82.02 | 71.74 | 81.18 | 70.06 |
| ViT-L/14-336px | 88.53 | 80.85 | 88.02 | 80.45 | 89.12 | 81.65 |
| ViT-L/14 | 88.79 | 81.44 | 88.57 | 81.01 | 89.73 | 82.60 |

## 5 CONCLUSION

In this paper, we propose SimE, a simple yet efficient incremental learning (IL) framework. SimE utilizes a pre-trained model as the encoder and incorporates adapters for finetuning, thereby achieving robust feature extraction capabilities while adapting to IL tasks without the need to store replay data. Our experiments demonstrate that SimE achieves competitive results, validating its effectiveness. To explore better methods of adapter connections, we introduce the Multi-Adapter and observe a remarkable phenomenon: there is not always a direct positive correlation between the number of adaptive adapter connections and the model's IL capabilities. Specifically, while increasing the number of adapter connections between transformer blocks positively impacts model performance, adding more adaptive connections within transformer blocks during small incremental steps does not enhance,and may even degrade the model's IL ability. Such improvements occur only at more advanced incremental stages. We also conducted a systematic study on CLIP and identified the optimal CLIP model for CIFAR-100 and TinyImageNet. Based on our findings, we recommend that SimE's backbone encoder utilize the image encoder from CLIP models pre-trained on larger datasets like LAION-2B and larger architectures such as ViT-L/14 for CIL tasks. In future work, we will explore combining SimE with different pre-trained large models and various types of adapters for other tasks.

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

# Supplementary Material

In the Supplement Material, we provide additional method details and experiment settings mentioned in the main text, as well as experiment results. The contents of supplementary material are organized as follows:

- **Section A** describes the update process of the encoder. The model starts with a pre-trained CLIP encoder, and after finetuning in the Adapter stage, a finetuned encoder is obtained. To maximize the feature extraction capability of the pre-trained model, we also include an additional pre-trained encoder. Together, these two encoders form the encoder of the model.

- **Section B** outlines the various experimental settings, including datasets, model backbone, evaluation metrics, CIL methods for comparison, and the training configurations of model. In the paper, unless otherwise specified, the experimental setup should be consistent with what is described here.

- **Section C** presents a comparison of our method with other CIL methods in terms of accuracy and efficiency, including accuracy comparisons across various tasks and between different datasets. This serves as a supplement to the experimental results in the main paper.

- **Section D** supplements the results of ablation experiments, including between transformer blocks and within blocks. In these experiments, "CLIP" represents the results when only the pre-trained encoder is used without an adapter.

- **Section E** supplement the comparison of SImE with state-of-the-art parameter-efficient methods across a wide range of datasets. Additionally, there are also visualize the representations of CLIP models pre-trained on different datasets via t-SNE.

## A    ADDITIONAL OPTIMISATION OF ADAPTER FOR DOMAIN ADAPTATION

The CLIP possesses exceptional zero-shot capabilities, enabling it to achieve performance comparable to supervised models on new datasets without finetuning. This underscores CLIP's powerful feature extraction abilities. However, CLIP still faces a domain gap between pre-trained datasets and downstream task datasets. For instance, while CLIP excels on datasets like ImageNet, its performance on MNIST(Radford et al., 2021) is poor. To bridge this gap, it is necessary to finetune CLIP for incremental learning downstream tasks. During the finetuning process, the weights $\theta_i, \phi_i$ of pre-trained CLIP image encoder $E'(\boldsymbol{c})$ in Eq.3 are frozen, and only the adapters and classifier are updated:

$$E^*(\boldsymbol{c}) = F(E'(\boldsymbol{c}), D), \qquad (9)$$

where $E^*(x)$ is the adapted CLIP image encoder, $F$ denotes the finetuning process, $D$ represents the data of the incremental tasks, $\eta$ refers to the trainable parameters, $\eta = \cup_j^Z \tilde{\eta_{ij}} \cup \theta_{Wt}$. Through finetuning, the pre-trained encoder can better adapt to downstream datasets. However, continuously finetuning on a series of tasks $D = \{D_1, \ldots, D_T\}$ would diminish its feature extraction capabilities due to Catastrophic Forgetting. Therefore, in this paper, we finetune the pre-trained encoder only on the first task $D_1$ to maximise the retention of the encoder's previous knowledge:

$$E^*(\boldsymbol{c}) = F(E'(\boldsymbol{c}_1), D_1). \qquad (10)$$

By finetuning on the first task, the encoder can better adapt to downstream datasets. However, the finetuning process will inevitably diminish the zero-shot capabilities of the pre-trained encoder. To better preserve the feature extraction capabilities of the pre-trained encoder, we concatenate the features output by the pre-trained encoder and the finetuned encoder and feed this combined feature set into the classifier for image classification.

$$E^c(\boldsymbol{c}) = \{E^*(\boldsymbol{c}); E(\boldsymbol{c})\} \qquad (11)$$

Here, $E^c(\boldsymbol{c})$ represents the composite encoder, and $\{\cdot; \cdot\}$ denotes the concatenation of the features output by the pre-trained encoder and the finetuned encoder. It is important to note that the concatenation is performed after finetuning, and encoder $E^c(\boldsymbol{c})$ will be frozen in subsequent tasks to maximise the retention of its feature extraction capabilities.

## B ADDITIONAL EXPERIMENTAL SETTING

**Datasets.** The experiments are implemented on CIFAR100 (Krizhevsky et al., 2009) and TinyImageNet (Zheng et al., 2023). CIFAR100 consist of 60K images with size of $32 \times 32$ from 100 classes, which are split on 2 classes, 5 classes and 10 classes in each step. Each class consist of 500 training and 100 testing samples. TinyImageNet, as a subset of ImageNet, consist of 100K images with size of $64 \times 64$ from 200 classes and each class consist of 500 training and 50 testing images. In our experiment, the 100 classes of TinyImageNet are split as base classes which are used for finetuning, and the rest 100 classes are split on 5 classes, 10 classes and 20 classes in each step.

**Network architectures in the SimE.** We have developed SimE, leveraging CLIP and adapters (Adaptformer (Chen et al., 2022) and Multi-Adapter) for class-incremental learning tasks. Within SimE, the CLIP image processor is utilized for data preprocessing. The image encoder, featuring various backbone sizes such as ViT-B/16, ViT-B/32, and ViT-L/14, is finetuned using adapters across different pre-trained datasets, including WIT-400B, Laion-400M, Laion-2B, Datacomp-1B, and CommonPool-1B. The classifier employs a fully connected (FC) layer, which uses class prototypes as weights.

**Evaluation metrics.** Following the methodology of Rebuffi et al. (2017), we assess SimE and compare it with other baseline methods using two metrics: Average Accuracy ($Avg_t$) and Last Accuracy ($Last_t$). ($Avg_t$) represents the mean of the Top-1 accuracy for every task, while ($Last_t$) denotes the Top-1 accuracy of final task. Mathematically, for the $t$-th task, Average Accuracy is calculated as follows: $Avg_t = \frac{1}{T} \sum_{t=1}^{T} Last_t$.

**Others IL methods.** We compare the SimE with existing CLIP-based methods (e.g., CoOp(Zhou et al., 2022), ZSCL(Zheng et al., 2023), Continual-CLIP(Thengane et al., 2022), AttriCLIPWang et al. (2023), and Boosting-CL(Yu et al., 2024)) and typical continual learning methods (e.g., LwF(Li & Hoiem, 2017), iCaRL(Rebuffi et al., 2017), DER(Yan et al., 2021), iTAML(Rajasegaran et al., 2020) and ARI(Wang et al., 2022a), UCIR(Hou et al., 2019), PASS(Zhu et al., 2021), and DyTox(Douillard et al., 2022)).

**Training procedures.** In this study, our experiments utilize the image encoder from CLIP(Radford et al., 2021). We finetune the SimE over 20 epochs on the first task for every datasets. Subsequently, all model weights, except the classifier, remain unchanged. During finetuning, we employ Stochastic Gradient Descent (SGD) as the optimizer. The starting learning rate is set at 0.01, adhering to a cosine decay schedule. We apply a weight decay of 0.0005, a batch size of 64, and the adapter's bottleneck dimension is set to 64.

## C ADDITIONAL COMPARISON ON ACCURACY

In this section, we present additional accuracy comparisons of different CIL methods and datasets. Tab.5 shows the Last accuracy for each task over 10 steps on CIFAR-100, compared with other CIL methods. We also compare the performance of our method on CIFAR-100 and TinyImageNet over 10 and 20 steps, as reported in Fig.6.

Table 5: Last accuracy of different CIL methods on CIFAR100. The accuracy of Task $t$, $t \in \{1, 2, ..., 10\}$ reported here is the last accuracy over all the previous tasks (i.e., Tasks $1, 2, ..., t$). If not otherwise specified, the method uses ResNet as the backbone, where † indicates the result based on the CLIP ViT-L/14 pre-trained on Laion-2B. The best results are coloured grey.

| Method | Task 1 | Task 2 | Task 3 | Task 4 | Task 5 | Task 6 | Task 7 | Task 8 | Task 9 | Task 10 |
|---|---|---|---|---|---|---|---|---|---|---|
| LwF(Li & Hoiem, 2017) | 89.3 | 70.1 | 54.3 | 45.8 | 39.8 | 36.1 | 31.7 | 28.9 | 24.4 | 23.9 |
| iCaRL(Rebuffi et al., 2017) | 88.7 | 78.1 | 72.4 | 67.2 | 63.7 | 60.2 | 56.4 | 54.4 | 51.9 | 49.5 |
| iTAML(Rajasegaran et al., 2020) | 89.2 | 89.0 | 87.3 | 86.2 | 84.3 | 82.1 | 80.7 | 79.1 | 78.4 | 77.8 |
| ARI(Wang et al., 2022a) | 88.6 | 86.9 | 85.8 | 84.6 | 83.1 | 81.8 | 81.6 | 81.0 | 80.2 | 80.9 |
| CoOp(W ViT-L/14)(Zhou et al., 2022) | 95.8 | 90.7 | 85.2 | 83.4 | 80.8 | 75.8 | 74.7 | 71.7 | 71.3 | 67.6 |
| Continual-CLIP(ViT-L/14)(Thengane et al., 2022) | 96.7 | 92.2 | 86.0 | 80.4 | 77.5 | 75.8 | 73.0 | 71.4 | 69.8 | 66.7 |
| AttriCLIP(ViT-L/14)(Wang et al., 2023) | 97.8 | 93.7 | 91.0 | 87.5 | 84.7 | 82.5 | 82.3 | 81.9 | 81.7 | 81.4 |
| **Ours(ViT-B/16 & AdaptMLP)** | 97.1 | 94.4 | 90.4 | 87.9 | 86.1 | 84.0 | 82.0 | 79.5 | 78.0 | 76.7 |
| **Ours(ViT-L/14 & AdaptMLP)** | 98.2 | 95.6 | 91.9 | 90.2 | 89.5 | 87.9 | 85.6 | 83.5 | 82.7 | 81.3 |
| **Ours(ViT-B/16 & AdaptAtten)** | 97.0 | 94.5 | 90.7 | 88.4 | 86.7 | 84.6 | 82.4 | 79.8 | 78.3 | 77.1 |
| **Ours(ViT-L/14 & AdaptAtten)** | 98.3 | 95.9 | 92.0 | 90.2 | 89.6 | 88.0 | 85.8 | 83.7 | 82.9 | 81.4 |
| **Ours**†$(ViT-L/14 \& AdaptAtten \& Laion-2B)$ | 98.2 | 96.9 | 94.6 | 93.1 | 92.5 | 91.1 | 89.4 | 87.7 | 87.2 | 86.0 |

Figure 6: Avg accuracy and Last accuracy of every Task on CIFAR100 and TinyImageNet with CLIP ViT-B/16. The Last accuracy of Task $t$, $t \in \{1, 2, ..., 20\}$ is the Top-1 accuracy over all the previous Tasks (i.e., Task $1, 2, ..., t$). and the Avg accuracy is the average value of Last accuracy over all the previous Tasks. The 100 classes of TinyImageNet are used as base classes.

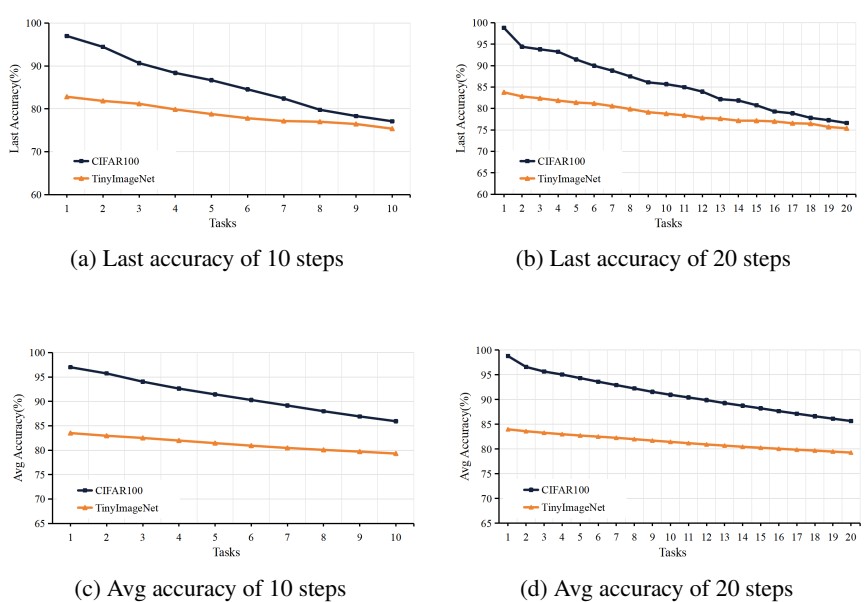

(a) Last accuracy of 10 steps        (b) Last accuracy of 20 steps

(c) Avg accuracy of 10 steps        (d) Avg accuracy of 20 steps

# D   ADDITIONAL INFLUENCE OF ADAPTER COMPONENTS IN SIME VIA ABLATION STUDIES

In this section, we report additional experiments on the influence of adapter components in SimE. We first study the influence of adapters connections between transformer blocks and report it in Tab6 & Tab.7, where "CLIP" indicates no adapter inserted in transformer blocks. ALL the ecperiments are conducted on CIFAR100 with CLIP ViT-B/16.

Table 6: Average accuracy of different continual learning methods on CIFAR100 with CLIP ViT-B/16. For example, "1-3" signifies that adapters are inserted in the first 3 blocks. The accuracy of Task $t$, $t \in \{1, 2, ..., 10\}$ reported here is the Top-1 accuracy over all the previous tasks (i.e., Tasks $1, 2, ..., t$). 'CLIP' means no adapter in model.

| Method | Task 1 | Task 2 | Task 3 | Task 4 | Task 5 | Task 6 | Task 7 | Task 8 | Task 9 | Task 10 |
|---|---|---|---|---|---|---|---|---|---|---|
| CLIP(Radford et al., 2021) | 91.6 | 89.6 | 85.3 | 82.6 | 80.48 | 78.13 | 75.17 | 72.65 | 71.3 | 70.08 |
| 1-3 | 95.60 | 93.55 | 89.77 | 87.50 | 85.70 | 83.48 | 81.49 | 79.35 | 77.77 | 76.47 |
| 1-6 | 95.80 | 93.20 | 89.90 | 87.45 | 85.84 | 83.87 | 81.94 | 79.66 | 78.04 | 76.65 |
| 1-9 | 97.00 | 93.95 | 89.87 | 87.40 | 85.64 | 83.30 | 81.17 | 78.91 | 77.28 | 76.00 |
| 1-12 | 97.10 | 94.35 | 90.37 | 87.90 | 86.10 | 84.02 | 82.00 | 79.49 | 78.00 | 76.70 |
| 4-6 | 95.20 | 92.90 | 89.43 | 87.00 | 85.50 | 83.48 | 81.61 | 79.34 | 77.63 | 76.26 |
| 4-9 | 96.60 | 93.80 | 89.73 | 87.15 | 85.30 | 82.90 | 80.89 | 78.62 | 77.16 | 75.60 |
| 4-12 | 96.80 | 94.05 | 90.13 | 87.48 | 85.66 | 83.43 | 81.37 | 78.97 | 77.39 | 75.93 |
| 7-9 | 95.40 | 92.90 | 89.13 | 86.28 | 84.22 | 81.82 | 79.57 | 76.81 | 75.17 | 73.72 |
| 7-12 | 95.40 | 93.25 | 89.60 | 86.55 | 84.66 | 82.25 | 80.06 | 77.26 | 75.81 | 74.31 |
| 10-12 | 94.20 | 92.00 | 88.53 | 85.65 | 83.62 | 81.15 | 78.74 | 76.11 | 74.53 | 73.18 |

Table 7: Effect of number and position of adapters loaded in CLIP image encoder. For example, "1-3" signifies that adapters are inserted in the first 3 blocks. All experiments are conducted on CIFAR100 with CLIP ViT-B/16, 'CLIP' means no adapter in model. The best results are coloured gray

| Blocks | 10 steps | | 20 steps | | 50 steps | |
|---|---|---|---|---|---|---|
| | Avg | Last | Avg | Last | Avg | Last |
| CLIP(Radford et al., 2021) | 79.69 | 70.08 | 80.41 | 70.08 | 80.80 | 70.08 |
| 1-3 | 85.07 | 76.47 | 84.76 | 75.66 | 83.10 | 72.86 |
| 1-6 | 85.23 | 76.65 | 85.40 | 75.84 | 83.86 | 73.50 |
| 1-9 | 85.05 | 76.00 | 85.26 | 75.70 | 84.01 | 73.66 |
| 1-12 | 85.60 | 76.70 | 85.30 | 76.02 | 84.09 | 73.77 |
| 4-6 | 84.84 | 76.26 | 85.08 | 75.58 | 82.30 | 71.59 |
| 4-9 | 84.78 | 75.60 | 84.60 | 74.65 | 82.94 | 72.31 |
| 4-12 | 85.12 | 75.93 | 84.81 | 75.03 | 83.09 | 72.46 |
| 7-9 | 83.50 | 73.72 | 83.40 | 73.26 | 81.79 | 71.14 |
| 7-12 | 83.91 | 74.31 | 83.50 | 73.54 | 81.96 | 71.42 |
| 10-12 | 82.77 | 73.18 | 82.86 | 72.89 | 81.66 | 71.16 |

We also investigate the influence of adapter connections within transformer blocks, as illustrated in Fig.7 &Tab.8 &Tab.9. Fig.7 presents the model performance on classes 70-100 during advanced steps (50 steps) with various Multi-Adapter implementations. Tab.8 report the results of different Multi-Adapter implementations across all steps, with the first row indicating the absence of an adapter in the encoder. Furthermore, we examine the influence of the bottleneck dimension of the adapter within the Multi-Adapter framework and report the results in Tab.9, with experiments conducted on Adapt-Atten.

Figure 7: The Last accuracy of different Multi-Adapter implementations on classes 70-100 in 50 step. Every step contains 2 classes, so numbers like "36" means classes 70-72. Experiments are conducted on CIFAR100 with CLIP ViT-B/16 and the classes are tested with the same sequence.

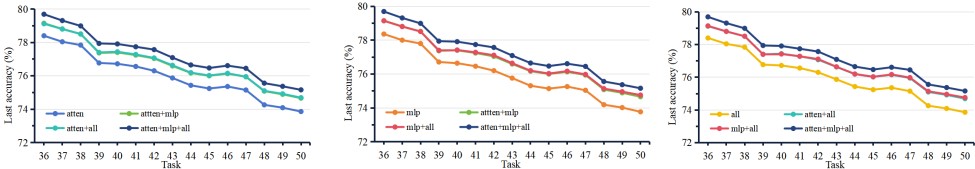

Table 8: The results of different implementations of Multi-Adapter with the bottleneck dimension being **1**. The structures of Adapter-MLP, Adapter-Atten and Adapt-All are shown in Fig2. "Para" refers to trainable parameters, with "M" standing for million. All experiments are conducted on CIFAR100 and the best results are coloured grey

| Adapt-MLP | Adapt-Atten | Adapt-All | Para(M) | 10 steps | | 20 steps | | 50 steps | |
|---|---|---|---|---|---|---|---|---|---|
| | | | | Avg | Last | Avg | Last | Avg | Last |
| ✗ | ✗ | ✗ | 0 | 79.69 | 70.08 | 80.41 | 70.08 | 80.80 | 70.08 |
| ✔ | ✗ | ✗ | 1.19 | 85.31 | 76.72 | 85.47 | 76.20 | 83.93 | 73.59 |
| ✗ | ✔ | ✗ | 1.19 | 85.94 | 77.61 | 85.33 | 75.98 | 83.80 | 73.50 |
| ✗ | ✗ | ✔ | 1.19 | 85.48 | 76.19 | 85.48 | 76.19 | 83.93 | 73.60 |
| ✔ | ✔ | ✗ | 2.38 | 85.98 | 77.41 | 85.67 | 76.42 | 84.60 | 74.54 |
| ✗ | ✔ | ✔ | 2.38 | 85.90 | 77.36 | 85.53 | 76.29 | 84.60 | 74.53 |
| ✔ | ✗ | ✔ | 2.38 | 85.76 | 77.20 | 85.31 | 75.90 | 84.51 | 74.50 |
| ✔ | ✔ | ✔ | 3.57 | 85.92 | 77.25 | 85.02 | 75.48 | 84.72 | 74.80 |

Table 9: The influence of bottleneck dimension of adapters. All experiments are conducted on CIFAR100 with CLIP ViT-B/16. "Bottleneck Dimension" means the projection dimension of adapters. Other experiment settings are the same as B

| Bottleneck Dimension | 10 steps | | 20 steps | |
|---|---|---|---|---|
| | **Avg** | **Last** | **Avg** | **Last** |
| 1 | 85.94 | 77.61 | 85.33 | 75.98 |
| 2 | 86.14 | 77.68 | 85.50 | 76.16 |
| 4 | 85.84 | 77.01 | 85.56 | 76.44 |
| 8 | 85.85 | 77.10 | 85.63 | 76.58 |
| 16 | 85.92 | 77.14 | 85.64 | 76.6 |
| 32 | 85.88 | 77.06 | 85.61 | 76.47 |
| 64 | 85.93 | 77.09 | 85.67 | 76.61 |
| 128 | 85.89 | 77.03 | 85.53 | 76.32 |
| 256 | 85.91 | 76.93 | 85.44 | 76.21 |

# E ADDITIONAL COMPARASION AND VISULIATION

we extend our experiments to include several SOAT parameter-efficient CIL methods that utilize pre-trained models on multiple datasets, including CIFAR-100, CUB-200, ImageNet-R(IN-R, ImageNet-100 (IN-100) and ImageNet 1000(IN-1K). The updated results are summarized in the table 10:

Table 10: Comparwssion between SimE and SOAT parameter-efficient CIL methods that utilize pre-trained models on multiple datasets. All experiments are conducted based on CLIP ViT-B/16.

| Methods | CIFAR 10 steps | | CUB 10 steps | | IN-R 10 steps | | IN-100 10 steps | | IN-1K 10steps | |
|---|---|---|---|---|---|---|---|---|---|---|
| | **Avg** | **Last** | **Avg** | **Last** | **Avg** | **Last** | **Avg** | **Last** | **Avg** | **Last** |
| L2PWang et al. (2020) | 81.90 | 73.08 | 71.90 | 62.99 | 81.67 | 75.98 | 80.51 | 67.22 | 79.30 | 69.60 |
| DualPromptWang et al. (2022b) | 81.45 | 72.51 | 71.74 | 62.14 | 82.01 | 75.77 | 80.65 | 67.38 | 79.39 | 69.79 |
| CODA-PromptSmith et al. (2023) | 76.98 | 62.25 | 66.61 | 50.88 | 78.00 | 67.52 | 64.13 | 34.76 | 76.99 | 66.96 |
| SLCAZhang et al. (2023) | 80.53 | 67.58 | 73.30 | 60.39 | 75.92 | 70.37 | 78.63 | 59.92 | 79.10 | 68.27 |
| APERZhou et al. (2023) | 75.76 | 65.50 | 78.80 | 70.61 | 78.62 | 71.35 | 85.84 | 76.40 | 76.60 | 68.74 |
| SimE(Ours) | **85.94** | **77.10** | **84.98** | **76.68** | **83.19** | **75.82** | **89.77** | **80.94** | **80.14** | **69.72** |

Figure 8: The t-SNE visualization of CLIP pre-trained on different datasets. All results are conducted on CIFAR100 with ViT-B/16 as backbone.

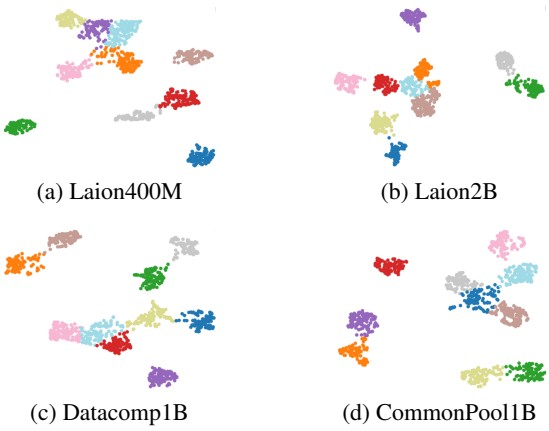

(a) Laion400M          (b) Laion2B

(c) Datacomp1B          (d) CommonPool1B

we also use t-SNE to visualize CLIP models trained on different datasets, as shown in Fig.8. It can be seen that the CLIP pre-trained on the larger dataset (LAION2B) clusters data points of the same class more tightly, indicating that it has a better ability to distinguish between different classes of data. Therefore, based on the t-SNE visualization results, we could selected CLIP models pre-trained on different datasets.

