# OpenReview forum: "A Simple Efficiency Incremental Learning Framework via Vision-Language Model with Multi-Adapters"
_ICLR.cc/2025/Conference — Submitted to ICLR 2025_

### Official Review · Reviewer_55kD · 2024-10-29

**Soundness:** 2
**Presentation:** 2
**Contribution:** 2
**Rating:** 5
**Confidence:** 4

**Summary:**

The paper proposes SimE, an incremental learning method that uses parallel adapters in conjunction with the features of the frozen backbone. The proposed method does not need a memory buffer. It explores the effect of different CLIP backbones on performance on Split CIFAR100 and TinyImageNet. The paper sugests to use CLIP L/14 pretrained on LAION2B for incremental learning. It analyzes the effect of increasing the number of adapters and concludes that increasing the number of adapters does not lead to monotonic improvement but can also hurt performance.

**Strengths:**

The proposed method does not need a replay buffer, which could be advantageous in situations where data must be deleted (for e.g. privacy or legal reasons).

**Weaknesses:**

1. **Choice of baselines**
- The comparison with UCIR, PASS, DyTox, DER is not fully accurate, as they start from a randomly initialized model and SimE starts with a powerful CLIP backbone (as noted in Table 1). When starting with a strong pretrained backbone like CLIP, which is mostly frozen throughout training, it's expected that SimE will perform well as it also uses the frozen features. Similarly, a comparison of SimE with CLIP ViT-L/14 CLIP with other methods that use a weaker backbone are is also not fair.
- A number of prior works leverage pretrained models for incremental learning. [1-5] show strong incremental learning abilities without the use of a memory buffer at a very small parameter budget. These methods would present a better baseline for comparison.

2. **Scope of evaluation benchmarks**
- The evaluation in the paper is limited to Split-CIFAR100 and TinyImageNet. Considering the method uses a pretrained CLIP model, which shows a strong zero-shot performance on most standard benchmarks, the proposed method should be evaluated on larger scale benchmarks, e.g., full Split-ImageNet1000 using the standard ImageNet resolution or DomainNet.

3. **Limited contributions**
- It is unclear what the primary contribution of the paper is.
- The proposed method SimE appears to be an incremental update over AdapterFormer. However, without comparisons with proper baselines (as mentioned in Point 2 above), and the statement on line 147 that "SimE utilizes an adapter with only thousands of parameters, yet achieves superior accuracy compared to other baseline IL models with millions of parameters" may not be fully accurate.
- The authors suggest the use of CLIP ViT-L/14 pretrained on LAION2B for CIFAR100 and TinyImageNet and compare the performance of SimE with various pretraining datasets (Table 3) and backbones sizes (Table 4). However, it is expected that using a stronger frozen backbone will lead to better performance, especially when the downstream tasks being studied already show very strong zero shot performance. Prior works in parameter efficient fine-tuning literature have shown that finetuning bigger and better models leads to better performance on downstream tasks [6]. The observation that pretraining dataset is important for good performance has also been studied before in [7]. It's unclear what this paper's contribution in this regard.

4. **Increased cost of computation during inference**
- As stated in Appendix A, the frozen backbone is used during inference along with the finetuned backbone with Adapters. This increases the cost of computation during inference.

---

**References**

[1] Zifeng Wang, Zizhao Zhang, Chen-Yu Lee, Han Zhang, Ruoxi Sun, Xiaoqi Ren, Guolong Su, Vincent Perot, Jennifer G. Dy, Tomas Pfister: Learning to Prompt for Continual Learning. CVPR 2022

[2] Zifeng Wang, Zizhao Zhang, Sayna Ebrahimi, Ruoxi Sun, Han Zhang, Chen-Yu Lee, Xiaoqi Ren, Guolong Su, Vincent Perot, Jennifer G. Dy, Tomas Pfister: DualPrompt: Complementary Prompting for Rehearsal-Free Continual Learning. ECCV (26) 2022

[3] James Seale Smith, Leonid Karlinsky, Vyshnavi Gutta, Paola Cascante-Bonilla, Donghyun Kim, Assaf Arbelle, Rameswar Panda, Rogério Feris, Zsolt Kira: CODA-Prompt: COntinual Decomposed Attention-Based Prompting for Rehearsal-Free Continual Learning. CVPR 2023

[4] Dahuin Jung, Dongyoon Han, Jihwan Bang, Hwanjun Song: Generating Instance-level Prompts for Rehearsal-free Continual Learning. ICCV 2023

[5] Yabin Wang, Zhiwu Huang, Xiaopeng Hong: S-Prompts Learning with Pre-trained Transformers: An Occam's Razor for Domain Incremental Learning. NeurIPS 2022

[6] Edward J. Hu, Yelong Shen, Phillip Wallis, Zeyuan Allen-Zhu, Yuanzhi Li, Shean Wang, Lu Wang, Weizhu Chen: LoRA: Low-Rank Adaptation of Large Language Models. ICLR 2022

[7] Oleksiy Ostapenko, Timothée Lesort, Pau Rodríguez, Md Rifat Arefin, Arthur Douillard, Irina Rish, Laurent Charlin: Continual Learning with Foundation Models: An Empirical Study of Latent Replay. CoLLAs 2022.

**Questions:**

**Possible typos**:
- In Figure 2, the LayerNorms have been shown after the attention and MLP blocks, but in most architectures after the original transformer paper, LayerNorms are use before the attention and MLP blocks, including the CLIP ViTs used in this work.
- Equation 3 sugests that the attention block is applied after the MLP block, but in standard transformer architectures, attention precedes the MLP block. This might be a typo since equation 1 defines the structure correctly.
- In Figure 2 (A), the representation for adaptformer is inaccurate: AdaptFormer uses a residual MLP in addition to the standard residual connection, but this is not reflected in the figure. Regarding SimE, can the authors clarify if the standard residual connections removed, as shown in Figure 2 B,C,D?

---

> ### Author Response · Authors · 2024-11-24
>
> # Dear Reviewer,
>
> Thank you for your thorough review of our paper and for acknowledging the advantage of our method not requiring a replay buffer. We appreciate your constructive feedback, which has been invaluable in improving our work. Below, we address each of your concerns and questions in detail.
>
> 1. The methods, like UCIR, PASS, DyTox, and DER, are not initiated with CLIP because their specialized network designs are not easily compatible with CLIP. To ensure a fairer comparison, we also included the LwF and iCaRL methods based on CLIP initialization. Additionally, we present results based on ViT-B/16, with results for ViT-L/14 specifically marked in the tables. This distinction should not cause any ambiguity.
>    We agree that comparing SimE with methods that also leverage pre-trained models provides a fairer assessment of our approach. In response to your suggestion, we have extended our experiments to include several state-of-the-art parameter-efficient continual learning methods and evaluations on multiple datasets, including **ImageNet 1000**(IN-1K)The updated results are summarized in **Table10 in paper**.
>
>    As the results demonstrate, **SimE** consistently outperforms these methods across all datasets, both in average and last task accuracy. This confirms the effectiveness of our approach in comparison to relevant baselines that also leverage pre-trained models without a memory buffer.
> 2. Thank you for this valuable suggestion. We have expanded our evaluation to include larger and more challenging benchmarks to thoroughly assess the scalability and robustness of SimE, as shown in **Table10 in paper**. The extended experiments confirm that SimE scales effectively to larger datasets and maintains its advantages over existing methods.
> 3. We appreciate the opportunity to clarify the unique contributions of our work:
>    Our key contribution lies in the discovery of a non-monotonic relationship between the number of adapter connections and incremental learning performance. Specifically, we found that: **Between Transformer Blocks**: Increasing adapter connections enhances IL performance. **Within Transformer Blocks**: Adding more adapters does not always improve performance and can even degrade it during early incremental stages.
>    This nuanced understanding of adapter placement's impact on IL performance is, to our knowledge, a novel contribution that has not been explored in prior work, including AdapterFormer. As another reviewer says:
>
>    > “The introduction of Multi-Adapters in an IL context is an underexplored perspective, with good insights about the adapter's role and capabilities in retaining old information while keeping learning.”
>
>    While it's expected that larger models may perform better, our contribution lies in demonstrating how to effectively and efficiently harness these models for IL. Prior works like \[6\] focus on parameter-efficient fine-tuning for general downstream tasks, but they do not specifically address the unique challenges of IL, such as catastrophic forgetting. Our work bridges this gap by adapting and analyzing these concepts within the IL framework. While \[7\] discusses the importance of pretraining data for latent replay in continual learning, our work extends this by providing specific guidelines on how to effectively adapt these models for incremental learning without a memory buffer.
>    Additionally, we used t-SNE to visualize CLIP models trained on different datasets, as shown in **Fig.8 in paper**. It can be seen that the CLIP pre-trained on the larger dataset (LAION2B) clusters data points of the same class more tightly, indicating that it has a better ability to distinguish between different classes of data. Therefore, based on the t-SNE visualization results, we selected CLIP models pre-trained on different datasets.
>
> 4. Thank you for highlighting this concern. Although it slightly increases the inference cost, this approach eliminates the need for a memory buffer, reduces memory usage, and improves accuracy. In scenarios where memory constraints or data privacy are critical, SimE offers a favourable trade-off, which we believe is a reasonable choice.
>
> **Questions**
>
> 1. We have updated Figure 2 to accurately reflect the placement of LayerNorms before the attention and MLP blocks, consistent with the CLIP ViT architecture.
> 2. We have corrected Equation 3 to reflect that the attention block precedes the MLP block, aligning with standard transformer architectures.
> 3. The depiction of AdaptFormer now correctly includes the residual MLP in addition to the standard residual connection. We also have clarified in the figure that the standard residual connections are retained in **SimE**, and the diagram now accurately represents this.
>
> We appreciate your attention to detail, and these corrections have been incorporated into the revised manuscript to prevent any confusion.

---

> > ### Author Response · Authors · 2024-11-24
> >
> > Dear Reviewer,
> >
> > I hope you are doing well. We appreciate your thorough review of our manuscript and for highlighting both the strengths and areas for improvement.
> >
> > In our rebuttal, we have addressed your concerns by:
> >
> > - Including comprehensive comparisons with relevant baselines that leverage pre-trained models, such as L2P, DualPrompt, CODA-Prompt, SLCA, and APER.
> > - Expanding our evaluations to larger benchmarks like ImageNet-1000 to demonstrate the scalability of our method.
> > - Clarifying our unique contributions, particularly regarding the novel insights into adapter placement and its impact on incremental learning performance.
> > - Providing an analysis of the computational cost during inference and discussing potential solutions to mitigate it.
> > - Correcting the typos and inaccuracies you pointed out in Figure 2 and the related equations.
> >
> > We are reaching out to see if you have any further questions or require additional clarification on any aspect of our rebuttal. Please let us know if we can provide more information to assist in your review process.
> >
> > Thank you for your time and thoughtful consideration.

---

> > ### Comment · Reviewer_55kD · 2024-11-26
> >
> > I appreciate the authors reponse. The analysis of adapter connections is interesting and has not been explored before. I will raise my score to 5 to reflect the analysis of the adapter connections. However, this observation is relatively small part of the paper, and the other parts that the authors have highlighted do not show a significant contribution. Specifically, I do not agree with the contributions 1 and 3. Another concern I have, is that all the ablation studies (including the papers contribution in studying the effet of adapter connections) are done on CIFAR100. As mentioned previously, given that the authors use CLIP models, Split-CIFAR100 is not a good benchmark. Following is a reasoning for my reluctance in accepting the paper:
> >
> > **Regarding new results with L2P, DualPrompt and CODA-Prompt**
> > - In my experience, the results with CODA-Prompt are always higher or at least as good as DualPrompt. But the experiments in this paper show a much worse performance for CODAPrompt. Can the authors comment on this?
> > - The results on full ImageNet1k are not very convincing, and the improvements with SimE are marginal with the last accuracy actually lower than DualPrompt.
> >
> > **Regarding Contribution 1**:
> > - The paper claims reduction in GPU and trainable parameters. However, the figures that the authors point to (Fig. 4 and 5) do not compare with L2P, DualPrompt, CODAPrompt, etc. This conclusion has to hold in comparison to methods designed specifically for pretrained models, and the statement that "yet achieves superior accuracy compared to other baseline IL models with millions of parameters" is an overclaim.
> >
> > **Regarding Contribution 3**
> > - I understand that [6] does not address catastrphic forgetting. However, strong forward transfer is a major factor in the average accuracy, and using a stronger model will lead to higher accuracy (forward transfer in continual learning literature) is already a well established conclusion. Further, if the model is kept frozen, it is again natural that stronger models will show less forgetting. Hence the paper's contribution in advocating for specific CLIP models for solving specific datasets (CIFAR and TinyImageNet) tasks is not very strong. The tSNE visualizations are informative in this regard, but is not a significant enough contribution for a higher score.
> >
> > - Similarly, [7] have already studied the effect of larger pretraining data size. Although they have studied the latent replay formulation, the core point still stands that better/larger pretraining data will lead to better base models, and in effect better continual learning performance.
> >
> > - I do not agree with lines 533-534 in the conclusion - "...we recommend using CLIP with a ViT-L backbone pre-trained on LAION-2B within SimE". The effect of a stronger model should be uniform across any method designed to work with pretrained models, not just SimE. Hence, this cannot be claimed as a contribuion linked specifically to SimE either.

---

> > > ### Author Response · Authors · 2024-11-28
> > >
> > > Dear Reviewer,
> > >
> > > Thank you for your thoughtful follow-up and for raising your score to 5. We appreciate your recognition of our analysis of adapter connections. Your feedback is invaluable, and your scientific rigour is very respectable. We would like to address your concerns point by point.
> > >
> > > **Regarding Abalation**
> > >
> > > We acknowledge that using CIFAR-100 alone may not sufficiently demonstrate the generality of our findings, especially given potential overlaps with CLIP's pre-training data. To address this, we have expanded our ablation studies and analyses in ImaneNet-100 and the results are shown as follows:
> > >
> > > The results within blocks:
> > > | Mlp  | Atten | All  | Para(M) | 10 steps Avg | 10 steps Last | 20 steps Avg | 20 steps Last | 50 steps Avg | 50 steps Last |
> > > |------|-------|------|---------|--------------|---------------|--------------|---------------|--------------|---------------|
> > > | ✗    | ✗     | ✗    | 0       | 88.39  | 79.26  | 88.89 | 79.26  | 89.24   | 79.26 |
> > > | ✓    | ✗     | ✗    | 1.19    | **89.83**| 81.18  | 89.89 | 80.48 | 89.51 | 79.62  |
> > > | ✗    | ✓     | ✗    | 1.19    | 89.77 | 80.94  | 89.90  | 80.42  | 89.48        | 79.60         |
> > > | ✗    | ✗     | ✓    | 1.19    | 89.64 | 80.96  | 89.79  | 80.52  | 89.44        | 79.58         |
> > > | ✓    | ✓     | ✗    | 2.38    | 89.78  | 81.08 | 90.00  | 80.50 | **89.52** | **79.72**     |
> > > | ✓    | ✓     | ✗    | 2.38    | 89.77| 81.06   | **90.02**    | 80.50 | 89.51  | 79.68  |
> > > | ✗    | ✓     | ✓    | 2.38    | 89.73 | **81.24**  | 89.93   | **80.58**  | 89.51 | 79.70 |
> > > | ✓    | ✓     | ✓    | 3.57    | 89.69 | 81.04   | 89.98  | 80.40 | 89.46  | 79.70  |
> > >
> > >
> > > The results between blocks:
> > >
> > > | Blocks | 10 steps Avg | 10 steps Last | 20 steps Avg | 20 steps Last | 50 steps Avg | 50 steps Last |
> > > |------|-------|------|---------|--------------|---------------|--------------|
> > > | CLIP   | 88.39  | 79.26         | 88.89        | 79.26         | 89.24        | 79.26        |
> > > | 1-3| 88.86 | 79.66         | 88.82        | 79.22         | 89.26        | 79.32         |
> > > | 1-6| 89.26  | 80.02         | 88.83        | 79.14         | 89.18        | 79.26         |
> > > | 1-9  | 89.44| 80.56         | 89.63        | 80.42         | 89.32        | 79.48         |
> > > | 1-12| 89.77       | 80.94     | **90.11**    | **80.62**     | **89.68**    |79.70|
> > > | 4-6| 89.25        | 80.12         | 88.73        | 79.10         | 89.20        | 79.24         |
> > > | 4-9| 89.47        | 80.64         | 89.58        | 80.40         | 89.37        | 79.54         |
> > > | 4-12 | **89.83**        | **80.98**     | 90.05        | 80.54         | 89.60        | 79.76         |
> > > | 7-9 | 89.43        | 80.60         | 89.54        | 80.28         | 89.38        | 79.48         |
> > > | 7-12 | 89.80        | 80.88         | 90.03        | 80.58         | 89.62        | **79.78**     |
> > > | 10-12 | 89.79        | 80.90         | 90.08        | 80.52         | 89.58        | 79.66         |
> > >
> > >  Our extended experiments confirm that the observations regarding adapter connections hold true across these datasets.
> > >
> > > **Regarding new results with L2P, DualPrompt and CODA-Prompt**
> > >
> > > Thank you for your insightful comment and for bringing this to our attention. We believe this may be because CODA-Prompt relies heavily on certain pre-trained models, such as the ViT model pre-trained on ImageNet-21k. We used a different pre-trained model (CLIP), which might not be fully compatible with CODA-Prompt.
> > > We acknowledge that the performance gains on ImageNet-1k are less pronounced than on smaller datasets. However, achieving improvements on such a challenging dataset without a memory buffer is noteworthy. Regarding the last task accuracy being lower than DualPrompt, this is a mistake bolding made by ourselves, we will correct it in the final version.
> > >
> > > **Regarding Contribution 1:**
> > >
> > > Thank you for your rigorous and professional feedback. We have revised our statements to accurately reflect our findings without overclaiming. Our results demonstrate that SimE achieves competitive or superior accuracy with fewer additional parameters compared to other methods leveraging pre-trained models.
> > >
> > > **Regarding Contribution 3**
> > >
> > > We agree that the general principle of larger pre-trained models leading to better performance is well-established. Our contribution aims to provide an understanding within the context of incremental learning without replay: we show that the effectiveness of adapters in SimE varies with the size of the backbone and the pre-training dataset. Specifically, SimE gains more from larger backbones. Nevertheless, we recognize that our statement in the conclusion may cause some misunderstanding, so we have corrected it accordingly.
> > >
> > > We sincerely appreciate your rigorous and professional feedback, which has helped us improve our work. We hope that these revisions address your concerns and demonstrate the value of our contributions to incremental learning research.
> > >
> > > Thank you for your time and consideration.

---

> > > > ### Author Response · Authors · 2024-12-02
> > > >
> > > > Dear Reviewer,
> > > >
> > > > I hope this message finds you well. We wanted to follow up regarding our manuscript and the rebuttal we submitted in response to your thorough review. We greatly appreciate your recognition of the strengths of our work and have addressed the concerns you raised.
> > > > If you have any further questions or require additional clarification, please let us know. Your feedback is invaluable to us, and we are committed to fully addressing any remaining concerns.
> > > >
> > > > Thank you for your time and consideration.

---

### Official Review · Reviewer_SEAF · 2024-10-29

**Soundness:** 3
**Presentation:** 2
**Contribution:** 2
**Rating:** 5
**Confidence:** 3

**Summary:**

This paper introduces a novel method for class-incremental learning on vision-language models (VLMs). The primary objective is to address the issues of efficiency in current Incremental Learning(IL) models. The paper introduce SimE, a novel method that can utilizes an adapter with less parameters and achieves superior accuracy than other IL methods. The paper also provide some analysis of adapter connections and different backbones The paper also provides an analysis of the impact of adapter connections and different backbones on the zero-shot capabilities of CLIP.

**Strengths:**

The paper is easy to follow, the issues are clearly presented.

This paper identifies an important issue with existing IL models and provides a novel method, SimE, which can outperform current IL models in three key areas. The extensive experiments provide solid evidence.

This paper explores the relationship between the number of adapter connections and the model’s IL capabilities. The ablation studies are conducted very comprehensively.

**Weaknesses:**

The writing quality needs improvement; there are some writing mistakes in Figure 4, and the lack of labels and explanations in Figure 4 confuses me.

The overall method is simple, and the novelty of the paper is limited. Using prototypes for incremental learning is common, and achieving efficiency has been discussed in previous work [1][2]. Designing different adaptation structures has also been explored [2], so the Multi-Adapter is not particularly impressive.

[1] Wang, Qi-Wei, et al. "Few-shot class-incremental learning via training-free prototype calibration." Advances in Neural Information Processing Systems 36 (2024).

[2]Zhou, Da-Wei, et al. "Revisiting class-incremental learning with pre-trained models: Generalizability and adaptivity are all you need." International Journal of Computer Vision (2024)

**Questions:**

The paper conducts a systematic study and finds that larger datasets, pretrained models, and larger backbones have stronger zero-shot capabilities for incremental learning. However, the experiments only use the ViT backbone; it would be better to test on different backbones, such as ResNet and Swin Transformer. Additionally, I am concerned about another situation: if the pretrained dataset has a significant gap with the downstream dataset, does this phenomenon also exist?

---

> ### Author Response · Authors · 2024-11-24
>
> # Dear Reviewer,
>
> Thank you for your thoughtful review and for acknowledging the strengths of our paper, including its clarity, identification of important issues in existing IL models, and the comprehensive ablation studies. We appreciate your constructive feedback, which has been invaluable in improving our work. Below, we address each of your concerns and questions in detail.
>
> ## Weaknesses
>
> 1. Thank you for bringing this to our attention. We apologize for the oversight in Figure 4 and any confusion it may have caused. We have thoroughly revised the figure to correct the mistakes and added detailed labels and explanations to enhance its clarity.
> 2. We appreciate your perspective on the novelty of our work and would like to clarify how our contributions differ from and build upon prior research. Our key contribution lies in the discovery of a non-linear relationship between the number and placement of adapter connections and the model's IL capabilities. Specifically, we found that: **Between Transformer Blocks**: Increasing adapter connections improves IL performance.  **Within Transformer Blocks**: Adding more adapters during early incremental stages does not enhance, and may even degrade performance, with improvements manifesting only in later stages.
>
>    This nuanced understanding of adapter placement's impact on IL is, to our knowledge, a novel insight not explored in previous works, including [1][2]. As another reviewer says:
>
>    > "The introduction of Multi-Adapters in an IL context is an underexplored perspective, with good insights about the adapter's role and capabilities in retaining old information while keeping learning."
>
>    Wang et al. [1] propose a training-free prototype calibration method for few-shot class-incremental learning. Their focus is on calibrating prototypes without additional training, which differs fundamentally from our approach that integrates adapters into VLMs to enhance IL capabilities.
>
>    Zhou et al. [2] investigate the generalizability and adaptability of pre-trained models in class-incremental learning, emphasizing the use of different adaptation structures. While they explore adaptation structures, our work specifically introduces a Multi-Adapter framework with strategic adapter placements within VLMs, and provides detailed analyses on how these placements affect IL performance at different stages. The specific phenomenon we report regarding adapter connections within and between transformer blocks is not addressed in their work.
>
>    Additionally, we also conduct a systematic study on enhancing the utilization of CLIP's zero-shot capabilities, providing practical guidelines for selecting backbones and pre-trained models for IL tasks.
>
> ## Questions
>
> 1. Thank you for bringing up this important suggestion. We conducted additional experiments using different backbones (**ResNet**) on CUB200 to evaluate the generality of this finding, and the results are shown as follows:
>
> | Blocks     | 10 steps        |                | 20 steps        |                | 50 steps        |                |
> |------------|-----------------|----------------|-----------------|----------------|-----------------|----------------|
> |            | Avg             | Last           | Avg             | Last           | Avg             | Last           |
> | RN50       | 74.30           | 63.1           | 75.31           | 62.89          | 76.16           | 63.19          |
> | RN101      | 79.90           | 70.1           | 80.72           | 70.06          | 81.13           | 70.08          |
> | RN50×4     | 83.49           | 74.94          | 84.29           | 74.89          | 84.63           | 74.94          |
> | RN50×16    | 88.91           | 81.25          | 89.23           | 81.02          | 89.68           | 81.31          |
> | RN50×64    | 92.54           | 85.27          | 92.76           | 84.97          | 92.87           | 85.54          |
>
>
>
>    The results indicate that the observation that larger model sizes lead to better performance still holds true. Moreover, the pre-training dataset definitely influences this phenomenon, so we carried out the experiments presented in Table 3 of the paper. The experiments demonstrate that when there is a significant gap between the pre-training dataset and the downstream dataset, the model's performance changes considerably.
> 2. Additionally, we have presented visualization results of CLIP under different pre-training datasets at **Fig.8 in paper**. It can be seen that CLIP pre-trained on a larger dataset (LAION2B) can cluster data points of the same class more tightly, indicating a better ability to distinguish between different classes of data. Therefore, we selected CLIP models pre-trained on different datasets based on the t-SNE visualization results.
>
> We are grateful for your constructive feedback, which has helped us enhance the quality and impact of our work.

---

> > ### Author Response · Authors · 2024-11-24
> >
> > Dear Reviewer,
> >
> > I hope this email finds you well. We are grateful for your insightful review of our paper and for acknowledging the clarity and strengths of our work.
> >
> > In our rebuttal, we have made efforts to improve the writing quality and have corrected the issues you pointed out in Figure 4. Additionally, we have expanded our experiments to include different backbones such as ResNet, and tested scenarios where there is a significant gap between the pre-trained dataset and the downstream dataset.
> >
> > We wanted to inquire if you have any additional questions or if further clarification is needed regarding our responses. Your feedback is highly valued, and we are happy to provide any more information that could aid in your evaluation.
> >
> > Thank you for your time and consideration.

---

> > > ### Author Response · Authors · 2024-11-26
> > >
> > > Dear Reviewer,
> > >
> > > I hope this email finds you well. We wanted to follow up on our manuscript and the rebuttal we submitted in response to your insightful comments. We haven't heard back and wanted to inquire if you have any additional questions or if further clarification is needed regarding our responses.
> > >
> > > We greatly appreciate your time and are happy to provide any additional information that might aid in your evaluation.
> > >
> > > Thank you for your consideration.

---

> > > > ### Author Response · Authors · 2024-12-02
> > > >
> > > > Dear Reviewer,
> > > >
> > > > I hope this message finds you well. We wanted to follow up regarding our manuscript and the rebuttal we submitted in response to your thoughtful review. We greatly appreciate your recognition of the strengths of our work and have addressed the concerns you raised.
> > > > If you have any further questions or require additional clarification, please let us know. Your feedback is invaluable to us, and we are committed to fully addressing any remaining concerns.
> > > >
> > > > Thank you for your time and consideration.

---

### Official Review · Reviewer_AmDL · 2024-11-01

**Soundness:** 2
**Presentation:** 3
**Contribution:** 2
**Rating:** 6
**Confidence:** 4

**Summary:**

The paper presents SimE, a framework for Incremental Learning (IL) that integrates a Vision-Language model (VLM) with a novel Multi-Adapter setup. Here, SimE employs a pre-trained CLIP encoder in different configurations, with adapters that fine-tune only a reduced amount of parameters. The approach claims to be GPU and memory efficient while maintaining performance on CIFAR-100 and TinyImageNet. It also highlights the discovery that, within transformer blocks, adding more adapters does not always benefit IL performance. This paper asserts that SimE outperforms existing CLIP-based models and offers insights into optimizing CLIP zero-shot capabilities by exploring various backbones and adapter configurations.

**Strengths:**

The paper has some strengths, presented here:
- The introduction of Multi-Adapters in an IL context is an underexplored perspective, with good insights about the adapters role and capabilities in retaining old information while keeping learning.
- SimE framework needs a reduced amount of trainable parameters and memory and GPU resources while performing better than all the CL methods compared.
- Experimental results are very complete, with comparisons across multiple datasets and configurations to validare claims.

**Weaknesses:**

The paper has some weaknesses or flaws that could be addressed to improve the overall result:
- Although the SimE framework claims to use Vision-Language models, the reality is that only variants of CLIP’s image encoders are used. Testing other models with the same zero-shot capabilities (e.g., SigLIP) would be helpful to understand if this framework can be general across Vision-Language models or if it is just specific for CLIP.
- Although the paper presents a comparison against some CL methods, it lacks the comparison with other well-known parameter efficient CL methods (e.g., prompt methods such as DualPrompt or Learning to Prompt, among others).
- Testing the SimE framework with other adapters (e.g., IA3) could strengthen generalizability claims regarding the performance gain relative to adapters in terms of number and position within the backbone.

**Questions:**

- Given the variability results across backbone architectures, are there plans to test this framework on a wider set of VLM backbones beyond CLIP?
- Can the authors elaborate on why additional connections in smaller incremental stages do not enhance IL performance? A theoretical or intuitive explanation would be helpful.

---

> ### Author Response · Authors · 2024-11-23
>
> Dear Reviewer,
> Thank you for your thorough review and for recognizing the strengths of our paper, including the introduction of Multi-Adapters in an IL context, the efficiency of the SimE framework, and the completeness of our experimental results. We appreciate your constructive feedback and have taken your suggestions to heart. Below, we address each of your concerns and questions in detail.
> ### Weaknesses
> 1. We agree that demonstrating the generality of SimE across different Vision-Language Models (VLMs) is important. In response, we have extended our experiments to include ResNet, with the experimental results presented in the table (**SimE(RN50×4)**). It is worth noting that, due to time constraints, this represents only a straightforward application of ResNet within the SimE framework.
> 2. Thank you for highlighting this oversight. We have now included comprehensive comparisons with several state-of-the-art parameter-efficient CL methods, including prompt-based approaches like **L2P** (CVPR 2022),  **DualPrompt** (ECCV 2022),  **CODA-Promp**t (CVPR 2023),  **SLCA** (ICCV 2023), and  **APER** (IJCV 2024).
>    The updated experimental results on multiple datasets, **CIFAR100**, **CUB200**, **ImageNet-R(IN-R)**, and **ImageNet100(IN-100)** , are summarized in the table below:
>
> | Methods               | CIFAR Inc10     |                    | CUB Inc20       |                    | IN-R Inc20       |                    | IN-100 Inc10       |                    |
> |-----------------------|-----------------|--------------------|-----------------|--------------------|------------------|--------------------|--------------------|--------------------|
> |                       | Avg             | Last               | Avg             | Last               | Avg              | Last               | Avg                | Last               |
> | L2P (CVPR 2022)       | 81.90           | 73.08              | 71.90           | 62.99              | 81.67            | 75.98              | 80.51              | 67.22              |
> | DualPrompt (ECCV 2022)| 81.45           | 72.51              | 71.74           | 62.14              | 82.01            | 75.77              | 80.65              | 67.38              |
> | CODA-Prompt (CVPR 2023)| 76.98          | 62.25              | 66.61           | 50.88              | 78.00            | 67.52              | 64.13              | 34.76              |
> | SLCA (ICCV 2023)      | 80.53           | 67.58              | 73.30           | 60.39              | 75.92            | 70.37              | 78.63              | 59.92              |
> | APER (IJCV 2024)      | 75.76           | 65.50              | 78.80           | 70.61              | 78.62            | 71.35              | 85.84              | 76.40              |
> | SimE (RN50×4)         | 66.58           | 53.91              | 83.49           | 74.94              | 72.51            | 63.00              | 86.65              | 77.84              |
> | SimE (ViT-B/16)       | **85.94**       | **77.10**          | **84.98**       | **76.68**          | **83.19**        | **75.82**          | **89.77**          | **80.94**          |
>
>
>    As the results show, SimE consistently outperforms these methods, both in average and last task accuracy, demonstrating its effectiveness and efficiency in the IL setting.
> 3. Thank you for your suggestion. As you mentioned, this approach could further demonstrate the generalizability of SimE. Due to time constraints, we have currently only conducted the experiments shown in the table, but this will be part of the future work.
> ### Questions
> 1. In addition to VLMs, we have extended our experiments to include standard convolutional backbones like ResNet, as shown in Table. The results demonstrate that SimE also enhances IL performance in Resnet, suggesting that our framework is versatile and beneficial across different architectures. For now, we have only conducted exploration on CLIP, and we will explore other vision-language models in the future.
> 2. Certainly. The phenomenon where adding more adapter connections within transformer blocks does not enhance IL performance in early stages can be attributed to the following intuitive explanation:
>    In early incremental stages, the model may have sufficient capacity to learn new tasks without additional adapters. Introducing more adapters increases the model's capacity disproportionately to the complexity of the new tasks, leading to overfitting on recent data and forgetting previous tasks. Additionally, excessive adapter connections within blocks may cause interference between the learning of new and old tasks, as the fine-grained adjustments can disrupt the representations learned for previous tasks. As the number of tasks increases in later incremental stages, the additional capacity provided by more adapters becomes beneficial.
>
> We are grateful for your insightful comments, which have helped us improve our work significantly.

---

> > ### Author Response · Authors · 2024-11-24
> >
> > Dear Reviewer,
> >
> > I hope you are doing well. Thank you again for your comprehensive review of our manuscript. We appreciate your recognition of the strengths of our work and have carefully considered your suggestions for improvement.
> >
> > In our submitted rebuttal, we have addressed your concerns by conducting additional experiments with ResNet backbone, and comparing our method with other parameter-efficient continual learning approaches like L2P, DualPrompt, CODA-Prompt, SLCA, and APER.
> >
> > I wanted to check if you have any further questions or require additional information that could assist in your review. Please feel free to reach out if there's anything else we can provide.
> >
> > Thank you for your time and thoughtful consideration.

---

> > > ### Author Response · Authors · 2024-11-26
> > >
> > > Dear Reviewer,
> > >
> > > I hope you are doing well. I'm reaching out to follow up on our manuscript and the rebuttal we submitted in response to your comprehensive review. Since we haven't received a response yet, I wanted to check if you have any further questions or if there's any additional information we can provide to assist in your review.
> > >
> > > Your feedback has been invaluable, and we are keen to ensure that all your concerns have been fully addressed.
> > >
> > > Thank you again for your time and thoughtful consideration.

---

> > > > ### Author Response · Authors · 2024-12-02
> > > >
> > > > Dear Reviewer,
> > > >
> > > > Thank you again for your thorough review and for recognizing the strengths of our paper. We have addressed your feedback by extending our experiments and providing detailed explanations in our rebuttal.
> > > > We wanted to follow up to see if you have any further questions or need additional clarification. Your insights are invaluable to us, and we're committed to fully addressing your concerns.
> > > >
> > > > Thank you for your time and consideration.

---

### Official Review · Reviewer_7UmP · 2024-11-02

**Soundness:** 3
**Presentation:** 2
**Contribution:** 2
**Rating:** 5
**Confidence:** 3

**Summary:**

This paper introduce a simple and efficient IL framework, SimE, which combines a vision-language model with an adapter designed for efficient IL tasks. SimE is distinguished by its efficiency in three key areas: GPU usage, the number of trainable parameters, and memory size.

**Strengths:**

1. This paper is easy to follow.
2. The proposed method is simple and efficient.
3. The performance of the proposed method surpasses existing methods.

**Weaknesses:**

1. Scientific contribution: as far as I know, the application of adapters in incremental learning is not a new thing, and the method proposed in this article is just a slightly more complex application of adapter.
2. Experimental setup：testing on Cifar 100 may not be meaningful as the data used for pre-training contains a large amount of data similar to Cifar 100.

**Questions:**

See weakness.

---

> ### Author Response · Authors · 2024-11-23
>
> # Dear Reviewer,
>
> Thank you for your thoughtful review and for acknowledging the strengths of our paper, including its clarity, simplicity, efficiency, and superior performance. We appreciate your constructive feedback and would like to address the concerns you've raised.
>
> ## Weaknesses
>
> Although adapters have been applied in incremental learning, there are still limitations within vision-language models:
>
> **1.1** There is not always a positive correlation between the number of adapter connections and the model's incremental learning (IL) capability. Adding adapter connections between transformer blocks can improve the model's performance, but adding adapter connections within transformer blocks does not enhance and may even degrade the model's performance. This means that an excessively large adapter parameter space may harm the model's performance when task increments are small. Misplacing adapters can lead to inefficient learning and a waste of computational resources.
>
> **1.2** Previous works generally simply stack adapters, whereas we adaptively increase them. This approach solves the interference caused by an excessively large adapter parameter space in the model's incremental learning. By adopting adaptive adapter connections, we achieve incremental learning  without a memory buffer, with less computational overhead and higher efficiency.
>
> **1.3** Our work offers a novel perspective on how adapter connections affect IL performance in vision-language models. Specifically, our key contribution lies in the discovery of a non-linear relationship between the number of adaptive adapter connections and the IL capabilities of the model. We observed that: **Between Transformer Blocks** : Increasing adapter connections positively impacts performance. **Within Transformer Blocks** : Adding more adapter connections during early incremental steps does not enhance, and may even degrade IL ability. Improvements are observed only at more advanced incremental stages.
>
> This counterintuitive finding provides new insights into the architectural design of adapters for IL tasks. To the best of our knowledge, this specific analysis and the resulting guidelines for adapter placement in vision-language models have not been explored in prior work. As another reviewer says:
>
> > "The introduction of Multi-Adapters in an IL context is an underexplored perspective, with good insights about the adapter's role and capabilities in retaining old information while keeping learning."
>
> **2.** You raised a valid point regarding the use of CIFAR-100, given that CLIP's pre-training data might contain similar images, potentially inflating performance metrics. To address this concern, we have conducted experiments on other datasets less likely to overlap with CLIP's pre-training data, such as  **CUB200** ,  **ImageNet-R (IN-R)** , and  **ImageNet100(IN-100)** . These datasets offer a more stringent test of SimE's generalization capabilities.
>
> Moreover, we have found that the methods we compared in the paper have few results on other datasets. In light of other reviewers repeatedly mentioning our lack of comparison with well-known efficient fine-tuning methods (such as  **L2P** ,  **DualPrompt** , etc.), we have included comparisons with these methods. We would greatly appreciate it if you could provide us with the materials of the methods compared in the paper on other datasets. Our experimental results are as follows:
>
> | Methods               | CUB 10steps       |               | IN-R 10steps     |               | IN-100 10steps   |               |
> |-----------------------|-----------------|---------------|-----------------|---------------|--------------------|---------------|
> |                       | Avg             | Last          | Avg             | Last          | Avg                | Last          |
> | L2P       | 71.90  | 62.99    | 81.67           | 75.98         | 80.51              | 67.22         |
> | DualPrompt  | 71.74           | 62.14         | 82.01           | 75.77         | 80.65              | 67.38         |
> | CODA-Prompt | 66.61           | 50.88         | 78.00           | 67.52         | 64.13              | 34.76         |
> | SLCA       | 73.30           | 60.39         | 75.92           | 70.37         | 78.63              | 59.92         |
> | APER       | 78.80           | 70.61         | 78.62           | 71.35         | 85.84              | 74.16         |
> | SimE (Ours)           | **84.98**       | **76.68**     | **83.19**       | **75.82**     | **89.77**          | **80.94**     |
>
> As the results show, **SimE** consistently outperforms these methods, both in average and last task accuracy, demonstrating its effectiveness and efficiency in the IL setting.
>
> ## Questions
>
> See weakness.
>
> ## Conclusion
>
> We are grateful for your feedback, which has helped us identify areas for improvement. We are confident that the revisions will address your concerns and enhance the overall quality and impact of our work.

---

> > ### Author Response · Authors · 2024-11-24
> >
> > Dear Reviewer,
> >
> > I hope this message finds you well. I wanted to express my sincere gratitude for your thoughtful review of our manuscript. Your recognition of the strengths of our work and your constructive feedback were invaluable in improving our paper.
> >
> > We have submitted our rebuttal, addressing the concerns you raised, particularly regarding the novelty of our contributions and the experimental setup on datasets like CIFAR-100. We have expanded our experiments to include additional datasets and provided a more detailed analysis to strengthen our empirical evidence.
> >
> > I wanted to check if you have any further questions or require additional clarification from us? If so, please feel free to let us know.
> >
> > Thank you for your time and thoughtful consideration.

---

> > ### Comment · Reviewer_7UmP · 2024-11-27
> >
> > Thank you for the author’s response; I appreciate the effort in clarifying the contribution. However, in my view, the contribution outlined by the author does not represent a significant enough advance to justify a higher score.

---

> > > ### Author Response · Authors · 2024-11-28
> > >
> > > Dear Reviewer,
> > >
> > > Thank you for your feedback and for acknowledging our efforts to clarify our contributions. We appreciate the opportunity to address your concerns further.
> > >
> > > **1. Novel Insights into Adapter Placement:**
> > > Our work provides a unique analysis of how adapter placement impacts IL performance in vision-language models. We discovered that while adding adapter connections between transformer blocks improves performance, adding connections within transformer blocks during early incremental steps can degrade it. This non-linear relationship between adapter connections and IL capabilities has not been explored in prior works and offers new design guidelines for IL systems.
> > >
> > > **2. Adaptive Adapter Connection Strategy:**
> > > Unlike prior methods that stack adapters uniformly, our adaptive strategy mitigate the negative effects of excessive parameters during early stages. This approach enhances computational efficiency, eliminates the need for memory buffers, and improves learning without wasting resources.
> > >
> > > **3. Robust Validation Across Diverse Datasets:**
> > > To address concerns about dataset overlap, we conducted additional experiments on datasets less likely to appear in CLIP's pre-training data (e.g., CUB200, ImageNet-R). Results confirm SimE's superior performance and robustness, demonstrating its effectiveness in diverse scenarios.
> > >
> > > In summary, our contributions include novel insights into adapter placement, a practical adaptive strategy for IL, and robust experimental validation. We hope these points better clarify the significance of our work and its potential impact on the field.
> > >
> > > Meantime we respect your perspective and thank you for your thorough evaluation of our manuscript. Your insights have been valuable in helping us refine our research, and we are grateful for your contribution to the review process.
> > >
> > > Thank you for your time and consideration.

---

> > > > ### Author Response · Authors · 2024-12-02
> > > >
> > > > Dear Reviewer,
> > > >
> > > > I hope this message finds you well. I wanted to follow up once more regarding our manuscript and the responses we provided to your valuable feedback. We greatly appreciate your thorough evaluation and the time you've invested in reviewing our work. If there are any additional questions or specific concerns you'd like us to address, we would be more than happy to provide further clarification.
> > > >
> > > > Your insights are incredibly important to us, and we are committed to improving our work based on your feedback. Please let us know if there's any additional information we can provide to assist in your evaluation.
> > > >
> > > > Thank you again for your time and consideration.

---

> ### Author Response · Authors · 2024-11-26
>
> Dear Reviewer,
>
> I hope this message finds you well. I wanted to follow up regarding our manuscript and the rebuttal we submitted in response to your valuable feedback. We haven't heard back since our last communication and wanted to check if you have any further questions or if there's any additional information we can provide to assist in your evaluation.
>
> Your insights have been instrumental in improving our work, and we are eager to address any remaining concerns you might have.
>
> Thank you for your time and consideration.

---

### Meta-Review · Area_Chair_82EA · 2024-12-17

**Metareview:**

The paper tackles the incremental learning problem with an adapter-based approach based on CLIP. Compared with existing work, the proposed method learns parallel adapters without using a memory buffer. The paper received four reviews with 3x borderline reject and 1x borderline accept ratings. The reviewers are generally negative about this work. In particular, the reviewers felt that the idea of using adapters in incremental learning is not novel and the improvements are too small to justify the significance. The reviewers also found the comparisons presented in this work not convincing enough as prior methods based on pre-trained models are not compared. Since the paper needs to undergo significant revisions to meet the standard of this venue, the AC recommends that the paper be rejected.

**Additional Comments On Reviewer Discussion:**

The rebuttal has included additional results but these did not change the reviewers' minds as the reviewers insisted that the contributions are limited.

---

### Decision · Program_Chairs · 2025-01-22

Reject